# Regulation of RUVBL1-RUVBL2 AAA-ATPases by the nonsense-mediated mRNA decay factor DHX34, as evidenced by Cryo-EM

Andres López-Perrote[1], Nele Hug[2], Ana González-Corpas[1], Carlos F Rodríguez[1], Marina Serna[1], Carmen García-Martín[1], Jasminka Boskovic[1], Rafael Fernandez-Leiro[1], Javier F Caceres[2], Oscar Llorca[1]*

[1]Structural Biology Programme, Spanish National Cancer Research Centre (CNIO), Madrid, Spain; [2]MRC Human Genetics Unit, Institute of Genetics and Molecular Medicine, University of Edinburghx, Edinburgh, United Kingdom

**Abstract** Nonsense-mediated mRNA decay (NMD) is a surveillance pathway that degrades aberrant mRNAs and also regulates the expression of a wide range of physiological transcripts. RUVBL1 and RUVBL2 AAA-ATPases form an hetero-hexameric ring that is part of several macromolecular complexes such as INO80, SWR1, and R2TP. Interestingly, RUVBL1-RUVBL2 ATPase activity is required for NMD activation by an unknown mechanism. Here, we show that DHX34, an RNA helicase regulating NMD initiation, directly interacts with RUVBL1-RUVBL2 in vitro and in cells. Cryo-EM reveals that DHX34 induces extensive changes in the N-termini of every RUVBL2 subunit in the complex, stabilizing a conformation that does not bind nucleotide and thereby down-regulates ATP hydrolysis of the complex. Using ATPase-deficient mutants, we find that DHX34 acts exclusively on the RUVBL2 subunits. We propose a model, where DHX34 acts to couple RUVBL1-RUVBL2 ATPase activity to the assembly of factors required to initiate the NMD response.

*For correspondence:
ollorca@cnio.es

**Competing interests:** The authors declare that no competing interests exist.

## Introduction

RUVBL1 and RUVBL2 are two closely related AAA-type ATPases that assemble as hetero-hexameric structures made of alternating subunits and comprising six ADP/ATP-binding domains (*Figure 1A*). RUVBL1 and RUVBL2 contain a unique domain II (DII) that protrudes from one side of the hexamer and defines two distinct faces of the ring, named ATPase-face and DII-face hereafter (*Cheung et al., 2010*; *Ewens et al., 2016*; *Gorynia et al., 2011*; *Lakomek et al., 2015*; *López-Perrote et al., 2012*). DII domains comprise an oligonucleotide/oligosaccharide-binding (OB) fold domain (DII external) that connects to the hexameric ring by a flexible region containing a ß-stalk and a helical bundle (DII internal). Each RUVBL1 and RUVBL2 subunit contains a nucleotide-binding pocket and the adjacent subunit in the hexamer provides the arginine finger motif required for hydrolysis. Several structures of RUVBL1-RUVBL2 complexes reveal that nucleotides are present in the complex even if they were not supplemented during purification (*Gorynia et al., 2011*; *Lakomek et al., 2015*; *Matias et al., 2006*; *Muñoz-Hernández et al., 2019*).

RUVBL1 and RUVBL2 are essential constituents of several large complexes. In various chromatin-remodeling complexes such as INO80 and SRCAP, these ATPases form a scaffold that organizes the architecture of other subunits in the complex (*Aramayo et al., 2018*; *Eustermann et al., 2018*; *Feng et al., 2018*). RUVBL1 and RUVBL2 also interact with RPAP3 and PIH1D1 proteins to form the R2TP complex, a HSP90 co-chaperone involved in the assembly and maturation of some large

**Figure 1.** RUVBL1-RUVBL2 interacts with DHX34. (A) Top panels: Schematic representation of RUVBL1 (blue) and RUVBL2 (pink) domains and catalytic motifs. Bottom panel: structure of the human RUVBL1-RUVBL2 hetero-hexameric ring with protruding domain II (DII), generated from atomic structures of RUVBL1 (PDB 2C9O) and RUVBL2 (PDB 6H7X) (*Matias et al., 2006*; *Silva et al., 2018*). Top and bottom views are shown with the color code from top panel. The ATPase-face and DII-face of the ring as well as the internal and external regions of DIIs are indicated. (B) Pull-down experiment testing the interaction of purified His-RUVBL1-RUVBL2 with DHX34, using His-tag affinity purification. Proteins bound to affinity beads were eluted and analyzed by SDS-PAGE and stained using Oriole Fluorescent Gel Stain (Bio-Rad). DHX34 was found to elute specifically only when His-RUVBL1-RUVBL2 was present. (C, D) Immunoprecipitation (IP) of transiently transfected HEK293T cells with T7-DHX34 from HEK293T cells was performed in the presence of RNase A. Inputs (0.5%) and anti-FLAG-IPs (20%) were subjected to western analysis using the indicated antibodies. Proteins bound to T7 tag affinity beads were eluted and analyzed by SDS-PAGE and western blot using antibodies against the T7 tag in DHX34 and RUVBL1 (C) or RUVBL2 (D). For Inputs (0.5%) and anti-T7 IP (20%) are shown. (E) IP experiment testing the interaction of FLAG-RUVBL1 and HA-RUVBL2 co-expressed in HEK293T cells with T7-DHX34. Inputs (0.5%) and anti-FLAG-IPs (20%) were analyzed by SDS-PAGE and western blot using antibodies against DHX34, RUVBL1 and RUVBL2. These antibodies detected both transfected and endogenous proteins and are indicated on the left site of the panel.

The online version of this article includes the following figure supplement(s) for figure 1:

**Figure supplement 1.** Testing the interaction between RUVBL1-RUVBL2 and NMD factors.

---

complexes including RNA polymerase II and members of the Phosphatidylinositol 3-kinase-related kinase (PIKK) family such as ATR, ATM, SMG1, and mTOR (*Houry et al., 2018*; *Martino et al., 2018*; *Maurizy et al., 2018*; *Muñoz-Hernández et al., 2019*; *Rivera-Calzada et al., 2017*). In all these complexes, the DII-face of the RUVBL1-RUVBL2 ring is used as scaffold platform for the interaction with other proteins, which are recruited by the DII domains. One exception is the C-terminal domain

of RPAP3 that binds at the ATPase-face of the ring through the interaction with RUVBL2 but not RUVBL1 (*Martino et al., 2018*; *Maurizy et al., 2018*; *Muñoz-Hernández et al., 2019*).

ATP binding or hydrolysis by RUVBL1 and/or RUVBL2 is essential to all the reported activities in cells (*Izumi et al., 2010*; *Rajendra et al., 2014*; *Venteicher et al., 2008*), but the purified proteins display very weak ATPase activity in vitro (*Nano et al., 2020*; *Gorynia et al., 2011*). Little is known about the function of RUVBL1-RUVBL2-mediated ATP hydrolysis or how this is regulated within the cell. Inhibition of the RUVBL1-RUVBL2 ATPase stabilizes interactions with clients and proteins involved in mTOR assembly in cells (*Yenerall et al., 2020*). This and the low rates of ATP hydrolysis indicates that RUVBL1-RUVBL2 might not function as a processive ATPase but rather as a switch regulated through interacting partners. Recent structures suggest that the interaction of proteins with the DII domains can alter the conformation of the RUVBL1-RUVBL2 hexameric ring. PIH1D1 in the R2TP complex and the insertion domain of Ino80 in the INO80 complex interact with the DII domains, inducing conformational changes in regions of the ATPase ring (*Aramayo et al., 2018*; *Muñoz-Hernández et al., 2019*). In the R2TP complex, the interaction of PIH1D1 also induces changes in a region at the N-terminus of one RUVBL2 subunit that contains two histidine residues (His[25] and His[27]) that contribute to the interaction with the nucleotide. However, it has not been demonstrated whether these changes could regulate the ATPase activity of RUVBL1-RUVBL2.

In addition to their role as components of some macromolecular complexes, RUVBL1 and RUVBL2 regulate several cellular processes, including the Fanconia Anemia (FA) pathway, the assembly of the telomerase holoenzyme, and nonsense-mediated mRNA decay (NMD), a mechanism that removes aberrant transcripts. In FA, RUVBL1 and RUVBL2 associate with some components of the pathway, and their depletion reduces the amount of the FA core complex (*Rajendra et al., 2014*). Interestingly, monoubiquitination of FANCD2 and FANCI, a measure of the activation of the FA pathway, can be rescued by siRNA-resistant versions of RUVBL1 but not by an ATPase-dead mutant. The assembly of the human telomerase holoenzyme also requires RUVBL1 and RUVBL2 and the levels of assembled telomerase in RUVBL1-depleted cells can be rescued by the expression of the wild-type protein but not by an ATPase-deficient mutant (*Venteicher et al., 2008*). An ATPase active RUVBL1-RUVBL2 complex is also required for phosphorylation of UPF1 by the SMG1 kinase, one of the key steps to initiate NMD (*Izumi et al., 2010*). In this work, we focus on the study of RUVBL1 and RUVBL2 ATPases in the context of the NMD pathway.

NMD controls the quality of mRNAs by removing aberrant transcripts containing premature termination codons (PTCs). These are generated as errors during transcription or splicing, but they also appear through germline mutations in a number of genetic disorders (*Hug et al., 2016*; *Kurosaki et al., 2019*). NMD also plays an important role in fine-tuning gene expression by regulating the abundance of a significant fraction of physiological transcripts (*Nasif et al., 2018*), affecting functions such as stem cell differentiation (*Lou et al., 2016*). NMD is initiated by the assembly of transient multi-subunit complexes containing the up-frameshift (UPF) core NMD factors, UPF1, UPF2, and UPF3b, bound to the target mRNA (*Hug et al., 2016*; *Kurosaki et al., 2019*). First, the RNA helicase UPF1 and its specific kinase SMG1 bind to eukaryotic release factors eRF1 and eRF3 at ribosomes stalled at a PTC, forming the so-called surveillance (SURF) complex, where SMG1 is inhibited by the interaction of SMG8 and SMG9, two factors that block the access to the kinase domain (*Gat et al., 2019*; *Melero et al., 2014*; *Zhu et al., 2019*). Remodeling of the SURF complex by the interaction with NMD core factors UPF2 and UPF3b bound to the Exon Junction Complex (EJC) leads to the phosphorylation of UPF1 by SMG1, which results in the recruitment of factors that promote mRNA degradation. A fully functional NMD response requires the contribution of several other proteins beyond the core NMD machinery, including DHX34 (*DEAH box protein 34*), a DEAH box family RNA helicase that is required to initiate NMD (*Hug and Cáceres, 2014*; *Longman et al., 2007*). DHX34 is made of a structural core containing two canonical recombinase A (RecA)-like domains, a winged-helix domain (WH), a helical bundle (the Ratchet domain), and an OB-fold domain, followed by a short C-terminal domain (CTD) (*Hug and Cáceres, 2014*; *Melero et al., 2016*). DHX34 promotes UPF1 phosphorylation and although the mechanism is unknown, current evidence suggests this occurring in complex with the SMG1 kinase and its substrate UPF1. A region around the RecA domains in DHX34 binds UPF1 directly, whereas the CTD interacts with SMG1 (*Hug and Cáceres, 2014*; *Melero et al., 2016*). DHX34 variants lacking the CTD domain fail to promote the degradation of a PTC-containing NMD reporter (*Melero et al., 2016*). Only recently, germline pathogenic variants of DHX34 were found in four families as specific to familiar forms of

acute myeloid leukemia (AML) and myelodysplastic syndrome (MDS) (*Rio-Machin et al., 2020*). Interestingly, the four variants reduced the recruitment of UPF2 and UPF3b to UPF1, resulting in reduced UPF1 phosphorylation, suggesting a link between DHX34, the NMD pathway and the etiology of some inherited forms of these myeloid malignances.

RUVBL1 and RUVBL2 are also potential new non-canonical NMD regulators. Knockdown of RUVBL1 or RUVBL2 reduces SMG1-mediated UPF1 phosphorylation and affects the degradation of a PTC-containing β-globin reporter. These defects could only be recovered with wild-type and not ATPase-dead mutant versions of RUVBL1 indicating that the ATPase activity of the RUVBL1-RUVBL2 complex is required to initiate the NMD response (*Izumi et al., 2010*). How RUVBL1 and RUVBL2 participate in NMD and how their ATPase activity is regulated in this pathway remains enigmatic.

Here, we demonstrate that RUVBL1-RUVBL2 hetero-hexamers can bind directly to DHX34 in vitro and in cells in culture. Cryo-electron microscopy (cryo-EM) reveals that DHX34 induces global conformational changes in the RUVBL1-RUVBL2 complex with the RUVBL2 subunit being mostly affected. DHX34 destabilizes the N-termini of all RUVBL2 subunits and in consequence ATP hydrolysis activity by RUVBL2 is reduced. This demonstrates that interacting partners can profoundly affect the structure and activity of RUVBL1-RUVBL2 hetero-hexameric complexes. Our results suggest that DHX34 could couple the regulation of RUVBL1-RUVBL2 ATPase activity to the assembly of the complexes that initiate the NMD response.

## Results

### RUVBL1-RUVBL2 and DHX34 form a complex in vitro and in cells

To uncover the function of RUVBL1 and RUVBL2 in NMD, we sought to address whether these AAA-ATPases can interact directly with some of the core protein factors involved in the initiation of NMD. Pull-down experiments from cell extracts cannot easily differentiate direct interactions from those associations that are mediated by connecting partners. Thus, we first tested which NMD factors could bind directly to RUVBL1-RUVBL2 using purified proteins and in vitro interaction assays, and subsequently verified them in cell lysates.

For this, RUVBL1 and RUVBL2 were co-expressed and purified as heteromeric complexes containing His-RUVBL1 and RUVBL2 in equimolar amounts, and forming oligomeric complexes where two hexameric rings interact though the DII domains, as revealed by cryo-EM (*Figure 1—figure supplement 1A*), and in agreement with what we described before (*López-Perrote et al., 2012*; *Martino et al., 2018*). We first tested the interaction between RUVBL1-RUVBL2 complexes and UPF1$^{115-914}$ (a truncated version lacking the flexible N- and C-terminal ends of UPF1), UPF2, UPF3b, EJC, and DHX34 by pull-down experiments using the His-tag in RUVBL1 (*Figure 1B*, *Figure 1—figure supplement 1B–D*). Of all these, only DHX34 interacted directly with RUVBL1-RUVBL2 forming a stable complex that has not been previously described. We then verified the in vitro results using immunoprecipitation experiments from cells. When DHX34 was pulled down from cells, both RUVBL1 and RUVBL2 were detected by western blot (*Figure 1C–D*). The interaction between DHX34, RUVBL1 and RUVBL2 in cell lysates was further confirmed in immunoprecipitation experiments using FLAG-RUVBL1 (*Figure 1E*). FLAG-RUVBL1 eluted also endogenous RUVBL1 and RUVBL2, in agreement with the oligomeric nature of the complexes formed by these ATPases. Together, these results reveal a direct interaction between RUVBL1 and RUVBL2 and DHX34.

We also analyzed the interaction with SMG1, the kinase that phosphorylates UPF1, since this is the key event triggering NMD and RUVBL1-RUVBL2 has been found in complexes containing SMG1 in cell extracts (*Izumi et al., 2010*). SMG1 was purified by affinity chromatography in complex with SMG8 and SMG9 and incubated with His-RUVBL1-RUVBL2. A pull-down experiment using the His-tag in RUVBL1 revealed that SMG1 forms a direct complex with RUVBL1-RUVBL2 and similar results were obtained in pull-down experiments using a FLAG-tag in SMG1 as bait. Curiously, in both types of experiments, SMG1 and RUVBL1-RUVBL2 interact but SMG8 and SMG9 were undetected in the elution, suggesting that the formation of a complex between RUVBL1-RUVBL2 and SMG1 exclude SMG8 and SMG9 (*Figure 1—figure supplement 1E–F*).

Together, these results indicate that RUVBL1-RUVBL2 can interact directly with the NMD factors SMG1 and DHX34. In this work, we focused on the characterization of how DHX34 binds to RUVBL1-RUVBL2 and the consequences of this interaction.

## Cryo-EM of the RUVBL1-RUVBL2-DHX34 complex

To gain deeper insight into the consequences of the interaction of DHX34 with RUVBL1 and RUVBL2, we purified the RUVBL1-RUVBL2-DHX34 complex for structural characterization. The purification was optimized so that yields and homogeneity of the complex were suitable for structural studies (*Figure 2A*). For this, we coupled the purification of DHX34 from HEK293 cells with its binding to RUVBL1-RUVBL2. FLAG-DHX34 from cell extracts was bound to an immunoaffinity anti-FLAG resin and the beads were washed and incubated with purified RUVBL1-RUVBL2 prior to elution. Next, we used cryo-EM to determine the structure of the RUVBL1-RUVBL2-DHX34 complex. Freshly purified RUVBL1-RUVBL2-DHX34 was applied to holey-carbon grids, vitrified and cryo-EM images were collected.

2D averages of RUVBL1-RUVBL2-DHX34 revealed hexameric RUVBL1-RUVBL2 complexes with putative density for DHX34 located at the DII-face of the ring. The interaction of DHX34 disrupts the RUVBL1-RUVBL2 dodecameric double-ring complexes, an effect observed before in the assembly of

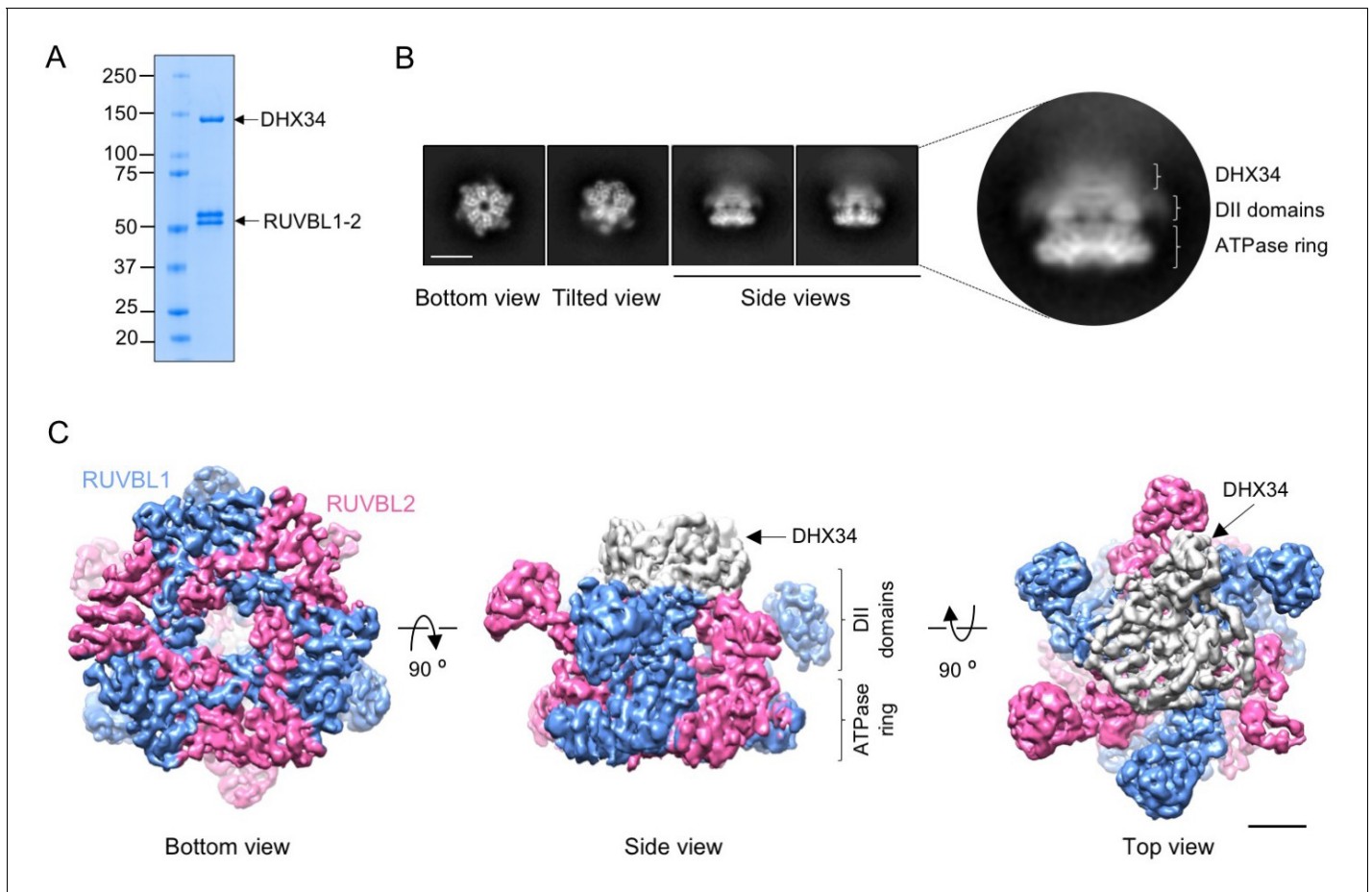

**Figure 2.** Cryo-EM of the RUVBL1-RUVBL2-DHX34 complex. (**A**) Purified RUVBL1-RUVBL2-DHX34 complex used for structural studies in a 4–15% SDS-PAGE stained with Quick Coomassie (Generon). (**B**) Representative reference-free 2D averages from cryo-EM images of the complex. Side views clearly show the projection of one ring with some density attached to the DII face (close-up right panel). Scale bar represents 10 nm. (**C**) Several views of the cryo-EM density obtained for RUVBL1-RUVBL2-DHX34 (RUVBL1 in blue, RUVBL2 in pink, and DHX34 in gray). ATPase core and DII domains in RUVBL1-RUVBL2 are indicated. Scale bar represents 25 Å.

The online version of this article includes the following video and figure supplement(s) for figure 2:

**Figure supplement 1.** Image processing of the cryo-EM images of the RUVBL1-RUVBL2-DHX34 complex workflow of the image processing strategy followed in this work.

**Figure supplement 2.** Resolution estimation of the cryo-EM for the RUVBL1-RUVBL2-DHX34 complex.

**Figure 2—video 1.** Cryo-EM structure of the RUVBL1-RUVBL2-DHX34 complex.

https://elifesciences.org/articles/63042#fig2video1

R2TP (*Martino et al., 2018*; *Muñoz-Hernández et al., 2019*; *Figure 2B*). An extensive 2D and 3D classification strategy of the images revealed that DHX34 attaches to the DII-face of the RUVBL1-RUVBL2 ring flexibly and we selected the most homogenous sub-group of particles for refinement (*Figure 2—figure supplement 1*, *Table 1*). The structure of RUVBL1-RUVBL2-DHX34 was determined at an average resolution of 5.0 Å (*Figure 2—figure supplement 2A*), with local resolutions up to 4.1 Å for the core RUVBL1-RUVBL2 whereas the resolution of DHX34 in the complex was around 8–10 Å (*Figure 2—figure supplement 2B* — *Figure 2—video 1*). The disparity in resolution between DHX34 and RUVBL1-RUVBL2 in the complex was a strong indication of the flexible attachment of DHX34 to the ATPases. Thus, the cryo-EM density for DHX34 could represent an average of several conformations if the hexameric RUVBL1-RUVBL2 directs alignment during image processing. Local resolution estimation for the OB folds in the DII domains of RUVBL1-RUVBL2 showed a similar resolution distribution (8–10 Å) due to their intrinsic flexibility (*Figure 2—figure supplement 2B*; *Martino et al., 2018*).

DHX34 binds to the internal regions of the DII domains in the hetero-hexameric RUVBL1-RUVBL2 ring. The DII external domains (OB-folds) are mostly free and potentially accessible for interaction with other partners (*Figure 2C*). DHX34 contacts several RUVBL1 and RUVBL2 subunits of the complex, thus inducing global conformational changes in the structure of the ATPases (see below).

## DHX34 induces conformational changes in the N-termini of RUVBL2

We first analyzed the structure of the RUVBL1-RUVBL2 ring using a mask that removed the influence of DHX34 and the protruding OB-fold domains during image processing (*Figure 2—figure supplement 1*). This way the average resolution of RUVBL1-RUVBL2 improved to 4.2 Å (*Figure 3A*, *Figure 3—figure supplement 1A,B*, *Table 1*) detecting the presence or absence of nucleotide in the nucleotide-binding pockets of both subunits (*Figure 3—figure supplement 1C,D*). The resolution was sufficient to model the structure of the RUVBL1-RUVBL2 ring after its interaction with DHX34 with the help of the crystal structure of RUVBL1-RUVBL2 (PDB 2XSZ) (*Figure 3—figure supplement 2A–C*, *Tables 2–3*).

The most striking feature in the cryo-EM map is that the N-terminal regions that contribute to nucleotide binding (*Muñoz-Hernández et al., 2019*; *Silva et al., 2018*) are visualized in RUVBL1 but not in any of the three RUVBL2 subunits (*Figure 3A,B*). We performed several experiments to fully verify that every RUVBL2 in the complex lacked density for the N-terminal region. We first searched for heterogeneity in RUVBL1-RUVBL2 by classifying the images of the ring in several 3D subgroups using a mask and we did not find sub-populations of particles where the N-terminal region of

**Table 1.** Cryo-EM data collection and parameters.
**Data collection and processing**

| Structure | RUVBL1-RUVBL2-DHX34 (EMD-11788) | RUVBL1-RUVBL2 ATPase core (EMD-11789) (PDB ID 7AHO) |
|---|---|---|
| Microscope | FEI Titan Krios | FEI Titan Krios |
| Detector | Gatan K2 (counting mode) | Gatan K2 (counting mode) |
| Magnification | 47756 | 47756 |
| Voltage (kV) | 300 | 300 |
| Electron exposure (e–/Å2) | 48.1 (40 fractions) | 48.1 (40 fractions) |
| Defocus range (µm) | −1.5 to −3.0 | −1.5 to −3.0 |
| Pixel size (Å) | 1.047 | 1.047 |
| Symmetry imposed | C1 | C1 |
| Initial particle images (no.) | 353 057 | 353 057 |
| Final particle images (no.) | 41237 | 101774 |
| Map resolution (Å) FSC threshold | 4.97 0.143 | 4.18 0.143 |
| Map resolution range (Å) | 4.0–12.0 | 3.8–6.0 |

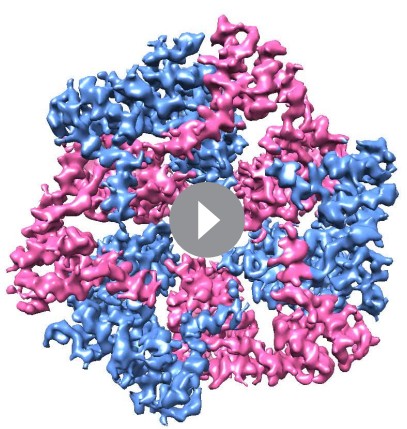

**Video 1.** Atomic structure of the RUVBL1-RUVBL2 ATPase ring. Several views of the atomic model for RUVBL1-RUVBL2 to show the conformational changes in the hexameric ring after the interaction with DHX34. During the video, the structure is superimposed to the atomic structure of the RUVBL1-RUVBL2 ATPase ring (PDB 2SXZ) in gray color, to highlight the differences in RUVBL2 and RUVBL1, and the loss of nucleotide in RUVBL2. In the final part of the video, the morphing between the crystal structure of the RUVBL1-RUVBL2 ATPase ring (PDB 2SXZ) and the structure described is this work highlights the conformational changes induced by DHX34 in the polypeptide chains. During morphing, nucleotides in both subunits and both conformations have been removed, as well as the N-terminal region of RUVBL2, which is only present in the crystal structure and it cannot be modeled in the DHX34-bound structure.

https://elifesciences.org/articles/63042#video1

RUVBL2 was visible (*Figure 3—figure supplement 3*). Next, we classified every position of RUVBL1-RUVBL2 dimers in six subclasses to search for heterogeneities within each position (*Figure 3—figure supplement 4*). More than 70% of particles in every position corresponded to particles clearly missing density of RUVBL2 N-terminus. Density for the N-terminus was not well defined for a small fraction of particles, but these corresponded to low-resolution structures due to the small number of images in these subgroups. Finally, we applied a symmetry expansion strategy as described before (*Martino et al., 2018*). For this, we rotated each particle around the 3-fold symmetry axis three times (0˚, 120˚, and 240˚) to place all RUVBL1-RUVBL2 dimers in the same position, thus triplicating the data set. Next, by placing a mask around one position now containing all existing RUVBL1-RUVBL2 dimers, we subjected the symmetry-expanded data to classification, searching for heterogeneity in all the available RUVBL1-RUVBL2 dimers regardless of its position in the ring (*Figure 3—figure supplement 5*). This analysis further confirmed that all dimers lack the RUVBL2 N-terminal regions. Based on these observations, we conclude that DHX34 induce changes that affect the conformation of RUVBL2 N-terminal regions in all three subunits of the complex.

## DHX34 induces the loss of nucleotide in every RUVBL2 subunit

The conformational changes in the N-terminus of RUVBL2 were coupled to the loss of nucleotide (*Figure 3C*). We and others have observed that RUVBL1-RUVBL2 co-purifies with nucleotides bound to all six subunits (*Gorynia et al., 2011*; *Lakomek et al., 2015*; *Matias et al., 2006*; *Muñoz-Hernández et al., 2019*), but whereas nucleotide remains bound to RUVBL1 in the RUVBL1-RUVBL2-DHX34 complex (*Figure 3D*), this is lost in all three RUVBL2 subunits (*Figure 3E*). The absence of density for nucleotide in RUVBL2 could not be attributed to a lack of sufficient resolution because nucleotides are clearly present in all RUVBL1 subunits in the same complex at similar resolution (*Figure 3—figure supplement 1C,D*).

DHX34 induces conformational changes in other regions of RUVBL1 and RUVBL2 besides the N-termini of RUVBL2 (*Figure 3—figure supplement 2B — Video 1*). These changes affect mostly to the internal regions of the DII domains and regions in the AAA-ring. These changes do not affect key residues of the RUVBL1 nucleotide-binding pocket, which explains why ADP remains bound to RUVBL1 after the interaction with DHX34 (*Figure 3D*). The structure of the nucleotide-binding pocket in RUVBL2 is not altered significantly with respect to the crystal structure of RUVBL2 bound to nucleotide (*Figure 3E*). This suggests that it is the displacement of the N-termini of RUVBL2 that is mostly responsible for the loss of nucleotide. Together, our results reveal that DHX34 affects nucleotide binding of every RUVBL2 subunit in the complex by stabilizing a conformation that displaces the N-termini of RUVBL2.

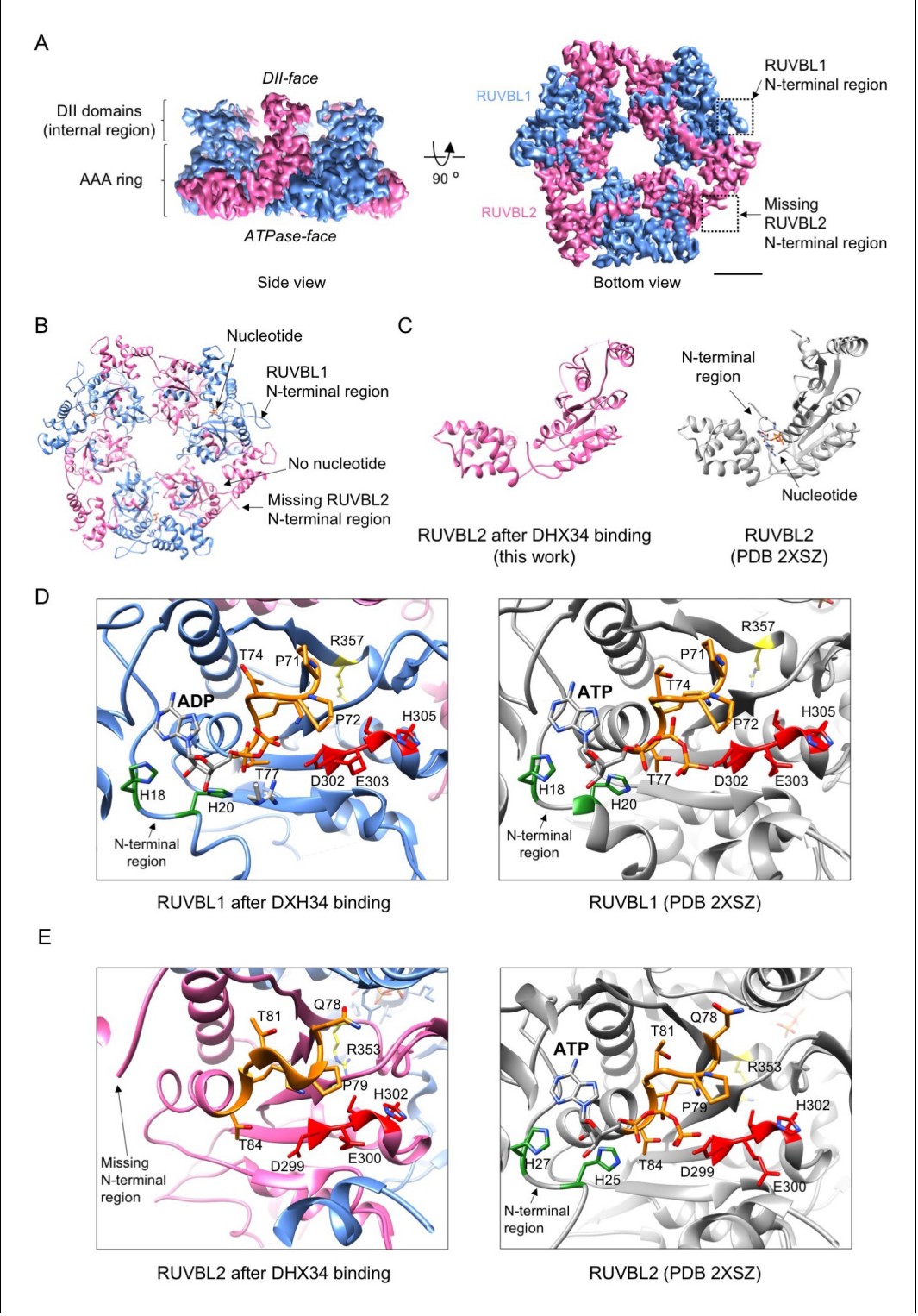

**Figure 3.** DHX34 induces large conformational changes in RUVBL2. (**A**) Side and bottom views of the RUVBL1-RUVBL2 ring obtained after refinement without the influence of DHX34 and the OB-fold domains. Squares highlight N-terminal segments of RUVBL1 (blue) and RUVBL2 (pink). Scale bar represents 25 Å. The presence and absence of RUVBL1 and RUVBL2 N-terminal regions is indicated only in one copy of each subunit, but it applied to all the subunits in the complex. (**B**) Bottom view of the atomic structure of RUVBL1-RUVBL2 ring modeled from the cryo-EM density. Color codes are as in (**A**). (**C**) Right panel: a view of the nucleotide binding region in RUVBL2 from the crystal structure of RUVBL1-RUVBL2 (PDB 2SXZ) in gray color; left panel: similar view of RUVBL2 in

*Figure 3 continued on next page*

*Figure 3 continued*

RUVBL1-RUVBL2 after DHX34 binding (this work, pink). (D) Close-up view of the nucleotide-binding regions in RUVBL1, comparing the structure after DHX34 binding (left panel) and the crystal structure of the RUVBL1-RUVBL2 complex (PDB 2SXZ) (right panel) in gray. N-terminal histidines (H18 and H20) are indicated in gray, Walker A residues in orange, Walker B in red, and the Arg finger in yellow. (E) As in (D) but for the RUVBL2 subunit. Color codes for relevant and catalytic motifs are represented as in (D).

The online version of this article includes the following figure supplement(s) for figure 3:

**Figure supplement 1.** Resolution estimation of the cryo-EM for RUVBL1-RUVBL2 after DHX34 binding.
**Figure supplement 2.** High-resolution features in cryo-EM map.
**Figure supplement 3.** Analysis of conformational changes in each RUVBL2 subunit.
**Figure supplement 4.** Analysis of conformational changes in each RUVBL2 subunit.
**Figure supplement 5.** Analysis of conformational changes in each RUVBL2 subunit.

## DHX34 makes multiple contacts through different domains with RUVBL1-RUVBL2

Next, we focused on the analysis of the structure of DHX34, whose resolution was lower than that of the RUVBL1-RUVBL2 hexameric ring. In an attempt to improve the resolution of DHX34, we extracted the density corresponding to the protein in each particle by applying density subtraction methods. These images were processed and classified without the influence of RUVBL1-RUVBL2 (*Figure 4—figure supplement 1A,B*). Despite our efforts, we were unable to improve the structure of DHX34. We suspect this could be a consequence of the limited number of images since small proteins need large datasets that allow classification and identification of the best particles in a

**Table 2.** Validation statistics for the atomic model of RUVBL1-RUVBL2 ATPase core.

**Refinement RUVBL1-RUVBL2 ATPase core**

| Software | phenix.real_space_refine Coot |
| --- | --- |
| Initial model used (PDB code) | 2XSZ |
| Model resolution (Å) | 4.1 |
| FSC threshold | 0.5 |
| Map sharpening *B* factor (Å2) | −222.98 |
| Model composition | 13521 |
| Non-hydrogen atoms | 1748 |
| Model composition | |
| Non-hydrogen atoms | 13521 |
| Protein residues | 1748 |
| Ligands | ADP (3) |
| R.m.s. deviations | |
| Bond lengths (Å) | 0.006 |
| Bond angles (°) | 0.911 |
| Validation | |
| MolProbity score | 1.70 |
| Clashscore | 6.95 |
| Poor rotamers (%) | |
| Ramachandran plot (%) | |
| Favored | 95.39 |
| Allowed | 4.61 |
| Disallowed | 0.00 |
| Rotamer outliers (%) | 0.21 |

**Table 3.** Correlation Coefficients (CC) of the atomic after model refinement.

Correlation Coefficients (CC) after model refinement of RUVBL1-RUVBL2_core map

| | |
|---|---|
| CC (mask) | 0.77 |
| CC (box) | 0.72 |
| CC (peaks) | 0.62 |
| CC (volume) | 0.76 |
| Mean CC for ligands | 0.84 |

homogenous conformation. This also indicates that the density of DHX34 of the cryo-EM map resolved in the context of the full RUVBL1-RUVBL2-DHX34 complex could be affected by the alignment of the AAA ring.

In isolation and at low resolution using negative-stain electron microscopy methods, DHX34 appears as a globular and compact protein comprising all of the conserved core domains, attached to a protruding extension corresponding to the C-terminal domain (CTD) (*Melero et al., 2016*; *Figure 4—figure supplement 1C*). The CTD domains appears thicker in the volume of DHX34 than in the electron microscopy images used to obtain the low-resolution structure due to its flexibility and the use of staining agents (*Melero et al., 2016*). We compared the cryo-EM volume of DHX34 extracted from the RUVBL1-RUVBL2-DHX34 complex with the low-resolution map of isolated DHX34. This comparison revealed that the dimensions of the DHX34 density that we observe attached to one ring of RUVBL1-RUVBL2 can only agree with one copy of DHX34, and that possibly, it is the DHX34 core domains and not the CTD that interact with RUVBL1-RUVBL2. Accordingly, purified DHX34 can still bind RUVBL1-RUVBL2 when its flexible C-terminal domain has been deleted (*Figure 4—figure supplement 1D*). In addition, we estimated that the approximate mass of the volume occupied by DHX34 in the complex with RUVBL1-RUVBL2 is around 120 kDa by using the *volume* programme in EMAN (*Ludtke et al., 1999*) (https://blake.bcm.edu/emanwiki/Volume). This tool estimates the mass enclosed by an isosurface using an average density for proteins of 1.35 g/ml (0.81 Da/A$^3$). Monomeric DHX34 and the DHX34 core domains alone have a molecular mass of 132 kDa and 111 kDa, respectively. Together, these analyses strongly suggest that one copy DHX34 interacts with one RUVBL1-RUVBL2 hexamer forming a complex with 3:3:1 stoichiometry.

Next, we tested whether any individual domain of DHX34 could have a significant role in mediating the interactions with RUVBL1-RUVBL2 in cultured cells. For this, we analyzed several DHX34 mutants (*Figure 4A*). We first tested a series of systematic domain deletion mutants and found that all single domain deletions retained binding to RUVBL1 and RUVBL2 to a measurable degree (*Figure 4B*). In order to test the importance of several DHX34 domains for RUVBL1 and RUVBL2 binding, we next analyzed larger truncated versions of DHX34 (*Figure 4C*). As shown by the quantification in *Figure 4D*, no domain is sufficient and necessary on its own to mediate the interaction with RUVBL1-RUVBL2. Heavily N- and C- terminal truncated versions of DHX34 showed a pronounced reduction in binding to RUVBL1-RUVBL2 but they could still bind to the ATPases (*Figure 4D*). Overall this analysis indicated that several interactions through different domains are made between DHX34 and RUVBL1-RUVBL2, which agrees with the structure of the complex that shows multiple regions of contact between DHX34 and the ATPases.

## DHX34 down-regulates RUVBL2 ATPase activity

We analyzed if the conformational changes induced by DHX34 correlate with changes in the ATPase activity of the His-RUVBL1-RUVBL2 complex (*Figure 5*). For these experiments, we expressed and purified the ATP-hydrolysis deficient His-RUVBL1$^{E303Q}$-RUVBL2$^{E300Q}$ mutant (*Gorynia et al., 2011*; *Lakomek et al., 2015*; *Matias et al., 2006*). We investigated whether these mutations affect the oligomerization of the complex using electron microscopy (*Figure 5—figure supplement 1A*). We found that His-RUVBL1$^{E303Q}$-RUVBL2$^{E300Q}$ forms similar oligomers to the wild-type complex and, in addition, it behaves similarly to His-RUVBL1-RUVBL2 when their intrinsic fluorescence was measured using a thermal denaturation assay (*Figure 5—figure supplement 1B*). In addition, we purified His-RUVBL1$^{E303Q}$-RUVBL2 and His-RUVBL1-RUVBL2$^{E300Q}$ containing the mutation in either RUVBL1 or RUVBL2. We also purified ATP-hydrolysis-dead mutant (DHX34$^{D279A}$) to remove the effects of ATP

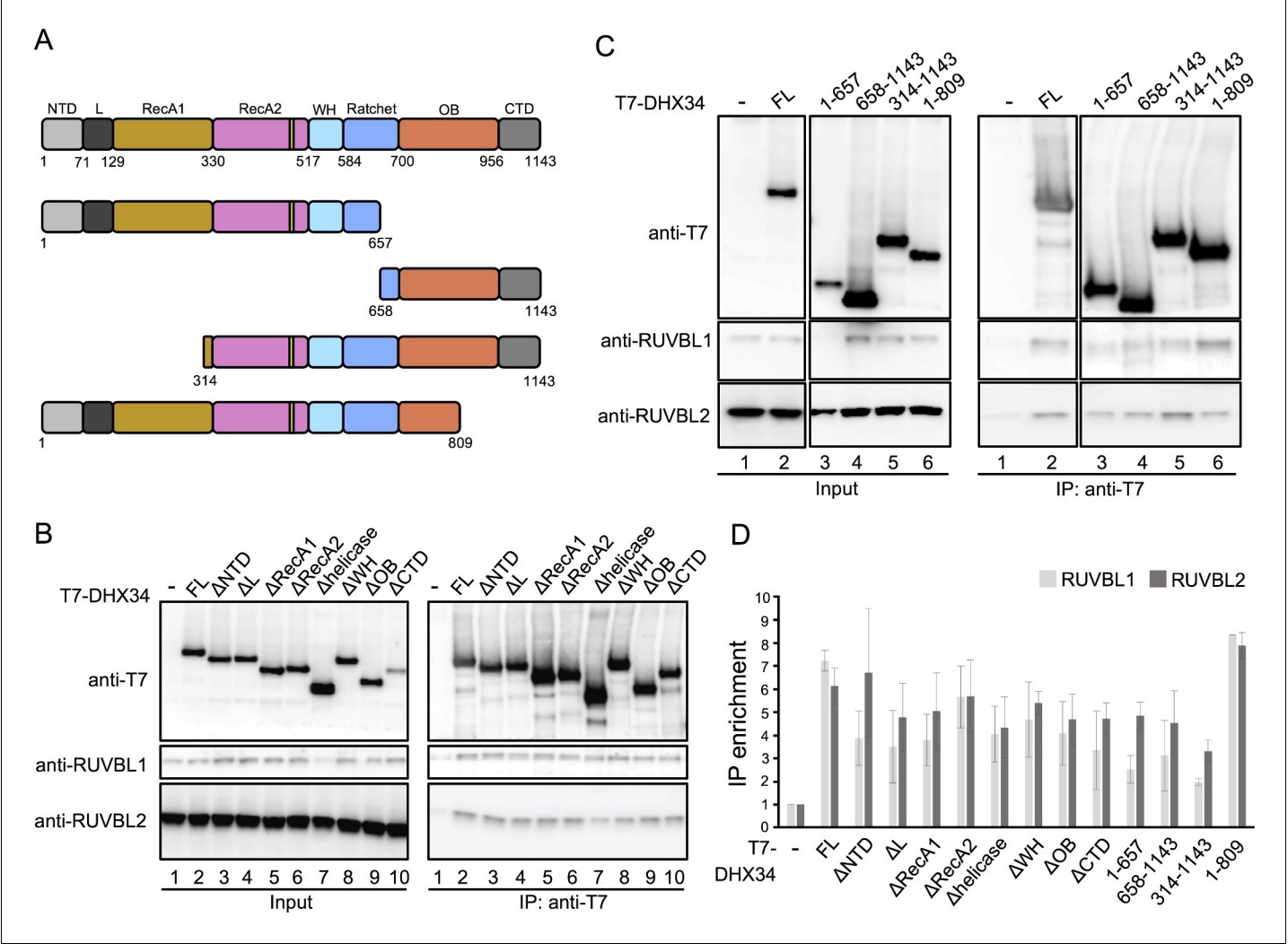

**Figure 4.** Mapping the interaction between DHX34 and RUVBL1-RUVBL2 in cells. (**A**) Cartoon depicting the functional domains of DHX34, showing the residue numbers that define their boundaries. Names of the domains are: N-terminal (NTD), L, RecA1, RecA2, winged-helix (WH), Ratchet, OB-fold and C-terminal (CTD). (**B–C**) Effect of domain deletions of DHX34 (**B**) and larger truncation including several domains (**C**) on the interaction with RUVBL1 and RUVBL2 using cell extracts. The Immunoprecipitation (IP) of T7-tagged versions of DHX34 was analyzed by western blot using antibodies against the T7 tag, RUVBL1 and RUVBL2. (**D**) Quantification of the experiments shown in 'B' and 'C'. The protein levels in the IP were quantified and normalized to the levels in the Input. Binding is expressed as IP enrichment compared to the empty vector control (-). For each expressed polypeptide, at least two independent experiments were analyzed.

The online version of this article includes the following figure supplement(s) for figure 4:

**Figure supplement 1.** Analysis of the images of DHX34 in complex with RUVBL1-RUVBL2.

hydrolysis by DHX34 in our experiments (*Hug and Cáceres, 2014*). DHX34[D279A] showed an ATP consumption of 24.2% compared with wild-type (*Figure 5—figure supplement 1C–E*). We also monitored DHX34 stability when incubated at 37°C, measured as changes in the intrinsic fluorescence during a thermal ramp. We found that DHX34 is not affected by 20 min incubation time but it is by 40 min when compared to freshly purified DHX34 (*Figure 5—figure supplement 1F*). This suggested that the protein might not be stable over long incubation times. Thus, ATPase measurements were restricted to 20 min using fresh preparations for which linearity of ATPase rates were maintained.

The ATPase activity of His-RUVBL1-RUVBL2 was determined using a spectrophotometric pyruvate kinase-lactate dehydrogenase-coupled assay that regenerates ATP so that the amount remains constant. The weak ATPase activity of His-RUVBL1-RUVBL2 that we measured at 37°C (4.9 ± 0.8 min$^{-1}$

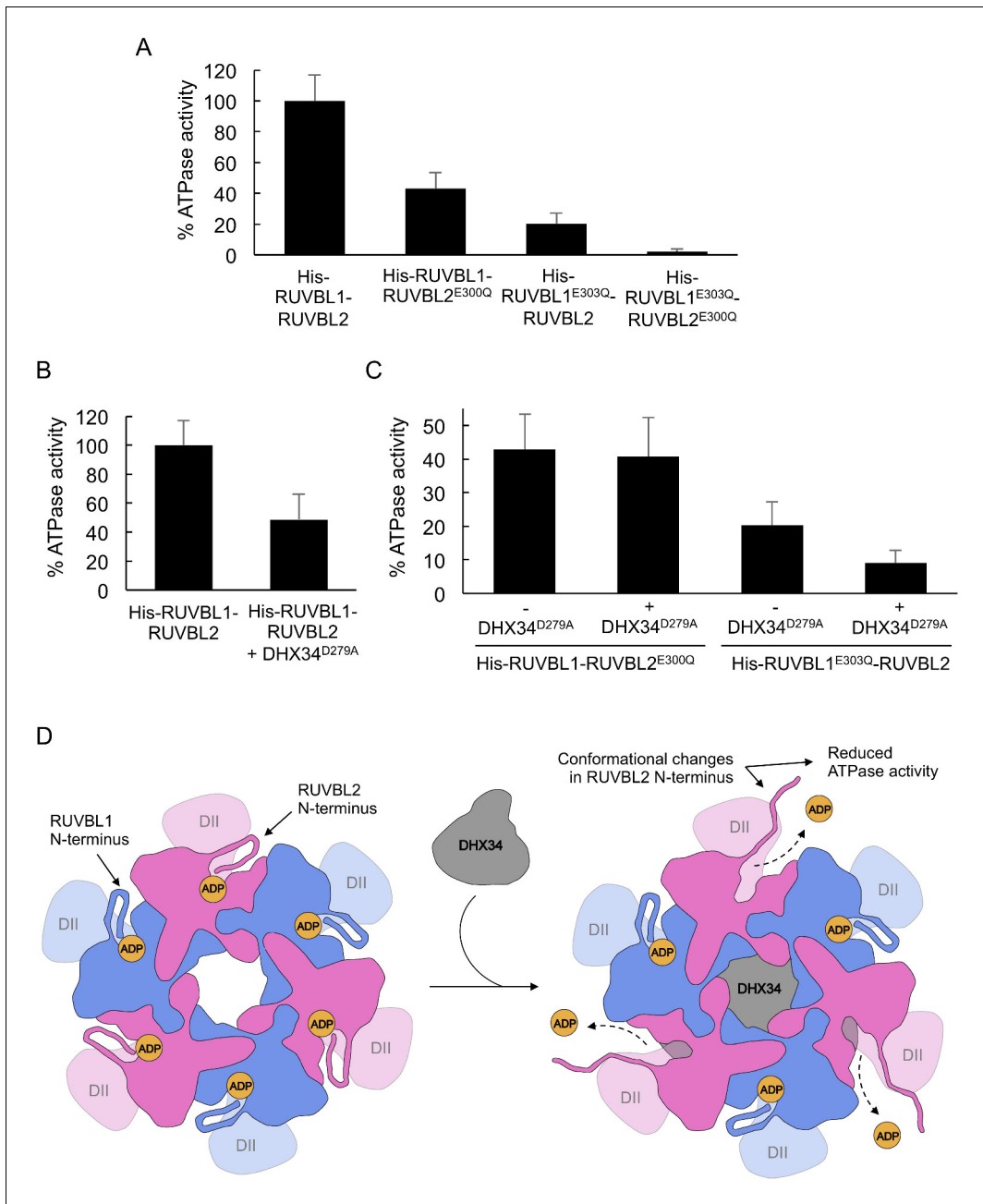

**Figure 5.** DHX34 down-regulates RUVBL2 ATPase activity. (**A**) Graph comparing the ATPase activity for His-RUVBL1-RUVBL2, His-RUVBL1$^{E303Q}$-RUVBL2, His-RUVBL1-RUVBL2$^{E300Q}$ and His-RUVBL1$^{E303Q}$-RUVBL2$^{E300Q}$ shown as percentage of the rate measured for wild-type RUVBL1-RUVBL2. Standard deviations from three independent experiments are indicated. (**B**) Graph comparing the ATP activity of His-RUVBL1-RUVBL2 in the presence and absence of DHX34$^{D279A}$, indicated as percentage of ATPase activity, assuming 100% activity for His-RUVBL1-RUVBL2. Standard deviations from four independent experiments are indicated. Sample His-RUVBL1-RUVBL2 used in these experiments contains a His-tag at the N-terminus of RUVBL1. (**C**) Graph showing the ATP activity for His-RUVBL1-RUVBL2 complexes where either RUVBL1 or RUVBL2 contains a mutation that abolished ATP hydrolysis (RUVBL1$^{303Q}$ or RUVBL1$^{E300Q}$) and the effect after incubation with DHX34$^{D279A}$. In all the experiments the ATPase activity for His-RUVBL1-RUVBL2 shown in 'B' is considered as 100%. Standard deviations from three independent experiments are indicated. (**D**) Model for the regulation of the ATPase activity of RUVBL1-RUVBL2 by DHX34. N-terminal regions in RUVBL1 (blue color) and RUVBL2 (pink color) subunits contribute interactions with the nucleotides in the nucleotide-binding pocket. DHX34 binds to the DII-face of the RUVBL1-RUVBL2 ring and induces large conformational changes in the N-termini of all RUVBL2 subunits promoting the loss of nucleotide

*Figure 5 continued on next page*

*Figure 5 continued*

and a down-regulation of the ATPase activity. Source files containing the data used for the time course measurements for ATP consumption in *Figure 5* and the figure supplement are available in *Figure 5—source data 1*.

The online version of this article includes the following source data and figure supplement(s) for figure 5:

**Source data 1.** Source data for the ATPase activity assays shown in *Figure 5*, *Figure 5—figure supplement 1*, *Figure 5—figure supplement 2* and *Figure 5—figure supplement 3*' and the caption to 'The file includes 15 sheets, each one for 1 sample, containing the replicas done for the sample.

**Figure supplement 1.** Oligomerization and stability of RUVBL1-RUVBL2 and DHX34 mutants.

**Figure supplement 2.** ATPase measurements and time courses for His-RUVBL1-RUVBL2 and several ATPase-dead mutants in the presence and absence of DHX34$^{D279A}$.

**Figure supplement 3.** DHX34$^{D279A}$ interferes with RUVBL1-RUVBL2 ATPase activity also after the His-tag in RUVBL1 was removed.

---

of ATP turnover) is similar to what had been determined previously by others (*Gorynia et al., 2011*; *Lakomek et al., 2015*; *Matias et al., 2006*; *Figure 5—figure supplement 2A*), and this value was set as 100% activity for comparison with subsequent measurements (*Figure 5A*). We verified that the activity measured corresponded to His-RUVBL1-RUVBL2 using His-RUVBL1$^{E303Q}$-RUVBL2$^{E300Q}$ (*Figure 5A*, *Figure 5—figure supplement 2A*). His-RUVBL1-RUVBL2$^{E300Q}$ and His-RUVBL1$^{E303Q}$-RUVBL2 complexes that are mutants in only one of the two subunits showed a significant reduction of their ATPase activity as expected if not all the subunits in each oligomer are active (43% and 20% ATPase activity for His-RUVBL1-RUVBL2$^{E300Q}$ and His-RUVBL1$^{E303Q}$-RUVBL2, respectively, compared to wild-type His-RUVBL1-RUVBL2) (*Figure 5A*, *Figure 5—figure supplement 2A*).

We then analyzed the consequences of adding DHX34 (2:1 DHX34:His-RUVBL1-RUVBL2 molar ratio, considering a His-RUVBL1-RUVBL2 hexamer and monomeric DHX34). Interestingly, the ATPase activity was reduced around 50% ($2.5 \pm 0.9$ min$^{-1}$) (*Figure 5B*, *Figure 5—figure supplement 2B*). Most interestingly, adding DHX34 caused no effects in His-RUVBL1-RUVBL2$^{E300Q}$ complexes containing ATPase-dead RUVBL2 subunits (His-RUVBL1-RUVBL2$^{E300Q}$ showed rates of $2.1 \pm 0.5$ and $2.0 \pm 0.5$ min$^{-1}$ in the absence and presence of DHX34$^{D279A}$, respectively) (*Figure 5C*, *Figure 5—figure supplement 2C*). In contrast, ATP hydrolysis by the His-RUVBL1$^{E303Q}$-RUVBL2 mutant was affected by DHX34 as in the wild-type (His-RUVBL1$^{E303Q}$-RUVBL2 showed rates of $1.0 \pm 0.3$ and $0.4 \pm 0.1$ min$^{-1}$ in the absence and presence of DHX34$^{D279A}$, respectively) (*Figure 5C*, *Figure 5—figure supplement 2D*). These results are fully in agreement with a model where DHX34 actions on the RUVBL1-RUVBL2 complex are mediated mostly by changes in the RUVBL2 subunits.

Finally, we verified that our results were independent of the presence or absence of the His-tag in RUVBL1 used to purify RUVBL1-RUVBL2. Removal of the tag had a small effect on the rate of ATP hydrolysis of the RUVBL1-RUVBL2 complex ($3.6 \pm 0.7$ min$^-$) (*Figure 5—figure supplement 3A,B*) but addition of DHX34 caused a similar reduction of ATP hydrolysis ($1.0 \pm 0.5$ min$^{-1}$), indicating that the presence of the tag does not interfere with the action of DHX34 (*Figure 5—figure supplement 3C, D*). As in the case of complexes containing His-RUVBL1, DHX34 caused a reduction in the ATPase activity of untagged RUVBL1-RUVBL2 complexes when RUVBL1 was mutated but not when RUVBL2 was mutated (Untagged RUVBL1-RUVBL2$^{E300Q}$ mutant consumed $2.0 \pm 0.5$ and $1.8 \pm 0.1$ min$^{-1}$ in the absence and presence of DHX34$^{D279A}$, respectively; and untagged RUVBL1$^{E303Q}$-RUVBL2 mutant showed rates of $1.1 \pm 0.5$ and $0.5 \pm 0.2$ min$^{-1}$ in the absence and presence of DHX34$^{D279A}$, respectively) (*Figure 5—figure supplement 3E–G*).

Together, these results indicate that there is a correlation between the conformational changes induced by DHX34 and the ATP hydrolysis activity of the complex. Thus, DHX34 regulates ATP hydrolysis of the RUVBL1-RUVBL2 complex mostly by changes in the RUVBL2 subunit.

## Discussion

Several lines of evidence show that the ATPase activity of RUVBL1 and RUVBL2 regulates several cellular processes. In particular, RUVBL1 and RUVBL2 ATPase activity is needed for a fully functional NMD response and for normal levels of SMG1-mediated UPF1 phosphorylation in vivo (*Izumi et al., 2010*; however, how these ATPases regulate NMD was completely unknown. Here, we provide

evidence that RUVBL1 and RUVBL2 directly interact with a subset of factors involved in the initiation of the NMD response and these interactions can affect greatly the conformation of RUVBL1-RUVBL2 oligomers and their ATPase activity. RUVBL1-RUVBL2 hetero-hexameric complexes interact directly with SMG1 and also with DHX34, two factors involved in regulating NMD initiation. SMG1 is the kinase that phosphorylates UPF1, one of the key events that dictates that an mRNA is targeted for degradation; *Kurosaki et al., 2019*). In this work, we focused on the study of the complex between RUVBL1-RUVBL2 and DHX34, a novel interaction that we find in vitro and in cells in culture. DHX34 is an RNA helicase that binds SMG1 and UPF1 and promotes the interaction of UPF1 with UPF2 and UPF3b, UPF1 phosphorylation and the initiation of NMD (*Hug and Cáceres, 2014*; *Melero et al., 2016*). Interestingly, variants of DHX34 unable to facilitate UPF1 phosphorylation have been found as pathogenic versions specific to inherited forms of acute myeloid leukemia (AML) and myelodysplastic syndrome (MDS) (*Rio-Machin et al., 2020*).

The regulation of RUVBL1-RUVBL2 ATPase activity is poorly understood. The N-termini of RUVBL1 and RUVBL2 contain two histidine residues, $His^{25}$ and $His^{27}$ in RUVBL2, that bind the nucleotide and contribute to maintain it within its binding pocket (*Muñoz-Hernández et al., 2019*; *Silva et al., 2018*). Here, cryo-EM reveals that the NMD factor DHX34 directly affects the conformation of RUVBL2 N-termini and decreases ATP hydrolysis. DHX34 distorts the N-terminal regions in all three RUVBL2 subunits that are not visible in the map, an indication that they are flexible and not attached to the hexameric ring after DHX34 binding. Thus, these changes provide an exit route for nucleotides, which are lost in all the RUVBL2 subunits (*Figure 5D*). We find that DHX34 down-regulates the ATPase activity of the RUVBL1-RUVBL2 complex in vitro. Importantly, these effects are mediated exclusively by changes in RUVBL2 and not in RUVBL1, as shown by RUVBL2 ATPase-dead mutants not being sensible to DHX34. Together, these results show that DHX34 stabilizes a conformation of RUVBL2 unable to hydrolyze ATP.

Despite conformational changes occurring in RUVBL1 and RUVBL2, major changes and loss of nucleotide is only observed in the RUVBL2 subunits. It remains unclear how this translates to the function of the hexameric ring, but this argues in favor of a model where RUVBL1 and RUVBL2 perform different functions in the complex. Some evidence supports this model. Several reports indicate that RUVBL1 and RUVBL2 do not always share the same function in vivo (*Mao and Houry, 2017*). The ATPase activity of RUVBL2 is several fold higher than RUVBL1 at least when expressed separately (*Nano et al., 2020*), suggesting they do not function in the same way. Along the same lines, RUVBL2 shows several specialized functions in the context of the R2TP complex. RUVBL2 binds RPAP3 and PIH1D1, two subunits of the R2TP complex, functions not shared by RUVBL1. Interestingly, cordycepin, a derivative of the nucleoside adenosine affects the circadian clock in mammals by targeting RUVBL2 (*Ju et al., 2020*). A crystal structure of RUVBL1-RUVBL2 bound to cordycepin (PDB 6K0R) shows that the compound interacted with all RUVBL2 subunits but not with RUVBL1 and the N-termini of RUVBL2 were visible and folded into the protein. Therefore, it seems that RUVBL1 and RUVBL2 subunits could have specialized functions within the complex, maybe also in the context of NMD.

The role of RUVBL1-RUVBL2 ATPase activity in cells is not understood beyond the fact that ATP binding and/or hydrolysis is essential in vivo for all the cellular pathways in which this has been analyzed. Thus, what the regulation of their ATPase activity could mean for NMD in cells is completely unknown. Purified RUVBL1-RUVBL2 complexes display low but measurable ATPase activity, which suggests that the complex might not work as a molecular motor but rather as a switch regulated by the interaction of partners. Recent experiments showed that RUVBL1 and RUVBL2 could pull-down partners of the R2TP chaperone pathway more efficiently when an inhibitor of their ATPase activity is used (*Yenerall et al., 2020*). This was interpreted as an indication that RUVBL1-RUVBL2 may form more stable complexes in the absence of ATP hydrolysis and that hydrolysis could serve to disengage RUVBL1-RUVBL2 from its partners. The initiation of the NMD response requires the assembly of several transient macromolecular complexes involving core and auxiliary NMD factors (*Hug et al., 2016*; *Kurosaki et al., 2019*). Only when the right set of interactions has taken place, SMG1 kinase phosphorylates UPF1, which triggers a transition from an initial surveillance complex (SURF) to a decay-inducing complex (DECID) and mRNA decay. DHX34 interacts with both SMG1 and its substrate UPF1 and somehow promotes interactions with other NMD factors that activate UPF1 phosphorylation (*Hug and Cáceres, 2014*; *Melero et al., 2016*). We can speculate that RUVBL1-RUVBL2 could contribute to these events by providing a platform that facilitates some of this complex set of

interactions, coupling their ATPase activity to the formation of some of the NMD complexes. The inhibition of RUVBL1-RUVBL2 ATP hydrolysis by DHX34 could maybe serve to stabilize this interaction while waiting for other NMD factors to bind DHX34 and RUVBL1-RUVBL2.

Together, our results reveal that DHX34, an NMD factor involved in NMD initiation, interacts directly with RUVBL1-RUVBL2 hetero-hexameric rings, profoundly modifying their structure and affecting their ATPase activity. This could help to couple RUVBL1-RUVBL2 ATPase activity to the assembly of the complexes required to initiate the NMD response.

## Materials and methods

### Transfections, immunoprecipitations, and western blotting

For interactions studies in cell extracts HEK293T cells were transfected with pcG-T7-DHX34 (WT and deletion mutants), pcDNA3-3xFLAG-UPF1 as described previously (*Hug and Cáceres, 2014*; *Melero et al., 2016*), pCDNA-3xHA-Reptin or pCDNA-3xFLAG-Pontin (*Izumi et al., 2010*; *Rajendra et al., 2014*; *Venteicher et al., 2008*) (Addgene plasmids), pcIneoFLAG-UPF2 (generous gift from Andreas Kulozik, Heidelberg) using Lipofectamine 2000 (Life Technologies) according to the manuals instructions. Cells were harvested and lysed 48 hr after transfection. Immunoprecipitation were performed, as previously described (*Hug and Cáceres, 2014*). Briefly, cells were lysed in IP buffer (10 mM Tris-HCl pH 8, 150 mM NaCl, 1 mM EGTA, 1% (v/v) NP-40, 0.2% (v/v) Na-Deoxycholate, Complete Protease Inhibitor (Roche), 1 mM dithiothreitol (DTT), 20 µg/ ml RNase A (ThermoScientific)). After Immunoprecipitations using T7 agarose (69026, MERCK Millipore), Anti-FLAG M2 Affinity Gel (A2220, Sigma-Aldrich) or antibodies against UPF1 (A300-038A, Bethyl) or UPF2 (sc-20227, Santa Cruz) coupled to protein G, immunoprecipitated proteins were separated by SDS-PAGE and detected by Western Blotting. RUVBL1 and RUVBL2 were detected with the following commercial antibodies: anti-Pontin (06–1299, Sigma Aldrich), anti-Reptin (SAB4200115, Sigma-Aldrich). FLAG and T7 affinity tags were detected with anti-FLAG (F3165, M2 clone, Sigma-Aldrich) and anti-T7 antibody (69522, Sigma Aldrich) respectively. The anti-DHX34 antibody has been previously described (*Hug and Cáceres, 2014*). Signals were detected with the ImageQuant LAS 4000 system (GE Healthcare) and quantified using the ImageQuant software.

### Cell culture

HEK293T cells were grown in high glucose Dulbecco's modified Eagle's medium (Life Technologies) supplemented with 10% (v/v) fetal calf serum (Life Technologies) and penicillin-streptomycin (Life Technologies) and incubated at 37°C in the presence of 5% $CO_2$.

### Cloning

For mapping experiments, a C-terminal His-tagged version of human RUVBL2 was produced. The cDNA of RUVBL2 (NM_006666) was PCR amplified from previously described pCDFDuet-1-RUVBL2 plasmid (*López-Perrote et al., 2012*) and inserted into pET21b vector (Novagen) using the IVA cloning system (*García-Nafría et al., 2016*), including 10 histidine residues at the C-terminus of the protein. RUVBL1 and RUVBL2 ATPase-dead mutants RUVBL1$^{E303Q}$ and RUVBL2$^{E300Q}$ unable to hydrolyze ATP were generated by site-directed mutagenesis of the original plasmids previously described (pETEV15b-RUVBL1 and pCDFDuet-1-RUVBL2) using standard protocols (*López-Perrote et al., 2012*). Oligonucleotides used for cloning are shown in *Table 4*.

### Expression and purification of recombinant proteins in bacteria

RUVBL2 including C-terminal His-tag (RUVBL2-His) was expressed in BL21 (DE3) *E. coli* cells (NZY-Tech) grown in LB medium. Expression of the protein was induced by addition of IPTG (Isopropyl β-D-1-thiogalactopyranoside) at a final concentration of 0.1 mM at 28°C for 4 hr when cells reached an optical density (OD) of 0.5. Cells were collected by centrifugation at 8000 rpm during 10 min at 4°C, and the pellet was resuspended in lysis buffer (50 mM Tris-HCl pH 7.4, 300 mM NaCl, 10% (v/v) glycerol, 0.1% (v/v) NP-40) supplemented with a cocktail of proteases inhibitors (cOmplete EDTA-free, Roche) and lysozyme (final concentration 0.1 mg/ml) (Sigma-Aldrich). Cells were lysed by sonication and clarified by centrifugation at 35,000 rpm for 1 hr at 4°C. Supernant containing soluble proteins was filtered using a 0.45 µm device and applied to a HisTrap HP affinity column (GE Healthcare)

**Table 4.** Oligonucleotides used for cloning.

| Construct | Name | Sequence 5´- 3´ |
| --- | --- | --- |
| pET21b-RUVBL2_H10 | pET21b_FW | CACCACCACCACCACCACTG |
| | RUVBL2_H10_FW | ATGGCAACCGTTACAGCCACTGTTTAACTTTAAGAAGGAGATATACAT |
| | RUVBL2_H10_RV | GGAGGTGTCCATGGTCTCGCGTGGTGGTGGTGGT GATGGTGATGGTGAGGTCCCTGGAACAGCACCTCCAG |
| | pET21b_RV | ATGTATATCTCCTTCTTAAAGTTAAACAAAATT |
| pETEV15b-RUVBL1_E303Q | R1_E303Q_FW | AGGTCCACATGCTGG |
| | R1_E303Q_RV | CATGTGGACCTGATCAACAAACAGCACACC |
| pCDFDuet-1-RUVBL2_E300Q | R2_E300Q_FW | AGGTCCACATGCTGGAC |
| | R2_E300Q_RV | GCATGTGGACCTGGTCGATGAACAGCACTCC |

equilibrated in buffer A (40 mM Tris-HCl pH 7.4, 200 mM NaCl, 10% (v/v) glycerol, 40 mM imidazole). Elution was performed using a gradient of increasing concentrations of imidazole with buffer B (40 mM Tris-HCl pH 7.4, 200 mM NaCl, 10% (v/v) glycerol, 500 mM imidazole). Fractions containing purified RUVBL2-His were pooled and dialyzed in buffer QA (40 mM Tris-HCl pH 7.4, 150 mM NaCl, 5% (v/v) glycerol) at 4°C during 16 hr. As a second purification step, dialyzed sample was applied on a HiTrap HP Q column (GE Healthcare) equilibrated in buffer QA and eluted in a gradient with increasing concentrations of NaCl using buffer QB (40 mM Tris-HCl pH 7.4, 1 M NaCl, 5% (v/v) glycerol). Fractions containing purified RUVBL2-His were dialyzed in buffer 40 mM Tris-HCl pH 7.4, 300 mM NaCl, 5% (v/v) glycerol at 4°C for 16 hr, freeze in liquid nitrogen, and storage at −80°C.

Expression and purification of His-RUVBL1 and His-RUVBL1-RUVBL2 complex were performed as previously described (*López-Perrote et al., 2012*). ATPase-dead mutants His-RUVBL1[E303Q]-RUVBL2, His-RUVBL1-RUVBL2[E300Q], and His-RUVBL1[E303Q]-RUVBL2[E300Q] were expressed and purified using the same protocol as for wild-type RUVBL1-RUVBL2. Untagged RUVBL1-RUVBL2 complexes including ATPase-dead mutants were obtained by TEV protease digestion as previously described (*López-Perrote et al., 2012*). Untagged UPF1[115-914] (UPF1 lacking residues 115–914 to increase protein stability), UPF2, UPF3b, and EJC (composed of eIF4AIII, Btz, MAGO and Y14 proteins) were produced as previously described (*Melero et al., 2012*). Chromatographic experiments were analyzed by 4–15% SDS-PAGE (MINI-PROTEAN TGX stain-free, Bio-Rad) and Quick Coomassie (Generon) staining.

## Expression and purification of recombinant proteins in mammalian cells

SMG1-SMG8-SMG9 complex was produced in HEK293T cells following previously published protocols, including tandem FLAG-SBP-HA tags at the N-terminus of SMG1, and Strep-HA tags at the N-terminus of SMG8 and SMG9. The SBP-tag at the N-terminus of SMG1 was used for purification by affinity chromatography as described before (*Melero et al., 2014*).

3xFLAG-DHX34, either wild-type, ATPase mutant D279A, or the deletion mutant DHX34_ΔCTD (lacking residues 956–1143) were expressed in suspension HEK293GnTi⁻ cells by transient transfection of the pCDNA vectors containing the different constructs of the DHX34 gene. Cells were grown in Freestyle 293 Expression Medium (Gibco) supplemented with 1% fetal bovine serum (FBS) in 5% $CO_2$ and 95% humidity with agitation. 24 hr prior transfection, 500 ml of cells were seed at $0.8·10^6$ cell/ml in 2L flasks. Transfection reagents were prepared by diluting 500 µg of the plasmids in Opti-MEM Reduced Serum Medium medium (Gibco), and after briefly mixing, 1500 µg PEI (Sigma-Aldrich) diluted in the same medium were added and further mixed. After 20 min incubation at room temperature, transfection mixtures were added to the cell cultures. Transfection was carried out for 48 hr in previously described grown conditions. Pellets for the transfected cells were collected by centrifugation at 2000 rpm for 10 min, washed twice with chilled PBS buffer, frozen in liquid nitrogen and stored at −80°C. For purification purposes, cells were resuspended in IP2 buffer (5 ml for 500 ml cell culture pellet) (10 mM Tris-HCl pH 8, 150 mM NaCl, 1 mM EGTA, 1% (v/v) NP-40) supplemented with Benzonase Nuclease (EMD-Millipore), proteases inhibitors (cOmplete EDTA-free, Roche) and phosphatases inhibitors (PhosSTOP EASYpack, Roche). Cell lysis was performed by freezing/thawing in liquid nitrogen three times, and lysate was clarified by centrifugation at 35,000 rpm for 1 hr at 4°C.

Supernatant was incubated with ANTI-FLAG M2 affinity resin (Sigma-Aldrich) previously equilibrated in IP2 buffer for 2 hr at 4°C with agitation. Using spin columns (SigmaPrep spin columns, Sigma-Aldrich), resin was subjected to washing steps with five resin volumes each: with IP2 buffer two times, IP 1M buffer (IP2 buffer with 1 M NaCl), F buffer (20 mM Tris-HCl pH 7.5, 150 mM NaCl, 1.2 mM EGTA, 250 mM sucrose, 1% (v/v) Triton X-100, 0.5% (v/v) NP-40 two times, F250 buffer) (F buffer supplemented with 250 mM LiCl), D buffer (20 mM HEPES pH 8, 100 mM KCl, 0.2 mM EDTA, 5% (v/v) glycerol, 0.5% (v/v) NP-40, and D400 buffer) (D buffer but with 400 mM KCl). Subsequently, the resin-bound protein was incubated with 5 mM ATP and 2 mM $MgCl_2$ in TBS buffer (50 mM Tris-HCl pH 7.4, 150 mM NaCl) for 20 min at room temperature in order to remove contaminating HSP70 (as identified by liquid chromatography-mass spectroscopy), and further washed two times with buffer TBS. Elution was performed for 20 min at room temperature with TBS buffer supplemented with 3xFLAG peptide (Sigma-Aldrich) at a final concentration of 0.15 µg/ml.

For the reconstitution of the RUVBL1-RUVBL2-DHX34 complex for structural studies, ANTI-FLAG M2 affinity resin loaded with 3xFLAG-DHX34 and after the washes, was incubated with a 6-fold molar excess of purified His-RUVBL1-RUVBL2 diluted in TBS buffer, further washed four times with TBS buffer, and eluted with 3xFLAG peptide as indicated for the 3xFLAG-DHX34 purification. In some preparations, endogenous HSP70 chaperone was detected as a contaminant in the elution of the complex, but this could be removed by incubating DHX34-loaded beads with ATP and $Mg^{2+}$ and subsequent washes in a buffer without nucleotide, prior to binding RUVBL1-RUVBL2.

## In vitro pull-down experiments

In vitro interaction assays were performed using pull-down experiments with purified proteins. For histidine affinity pull-down of His-RUVBL1-RUVBL2 and NMD factors, 7.5 µM of His-RUVBL1-RUVBL2 (containing a 10 histidine tag at the N-terminus of RUVBL1 and wild-type RUVBL2) were mixed with 2-fold molar excess of either UPF1[115-914], UPF2, UPF3b, EJC, SMG1-SMG8-SMG9 or DHX34 in 30 µl reactions in binding buffer (20 mM HEPES pH 7, 125 mM NaCl, 1 mM $MgCl_2$, 10 mM imidazole, 2.5% (v/v) glycerol, 0.1% (v/v) NP-40). After 15 min incubation at 4°C, Ni-NTA agarose resin (Qiagen) previously equilibrated in binding buffer was added to the mixtures and further incubated for 30 min at 4°C with agitation. Mixtures were included on centrifugation columns (SigmaPrep spin columns, Sigma-Aldrich), and unbound proteins were washed times with 10 resin volumes of binding buffer supplemented with 50 mM imidazole, followed by elution with binding buffer supplemented with 500 mM imidazole. Inputs (2 µl) and elutions (10 µl) samples were analyzed by 4–15% SDS-PAGE (MINI-PROTEAN TGX stain-free, Bio-Rad) and staining with Quick Coomassie (Generon) or Oriole Fluorescent Gel Stain (Bio-Rad). Similar protocols were used in interactions assays between His-RUVBL1-RUVBL2 complex and 3xFLAG-DHX34_ΔCTD truncated mutant, and between either His-RUVBL1 (10 histidines tag at N-terminus) or RUVBL2-His (10 histidines tag at C-terminus) and 3xFLAG-DHX34.

For immunoaffinity pull-down experiments using FLAG or 3xFLAG tagged proteins, similar reaction mixtures were prepared for FLAG-SMG1-SMG8-SMG9 or 3xFLAG-DHX34 and His-RUVBL1-RUVBL2 complex. ANTI-FLAG M2 affinity resin (Sigma-Aldrich) previously equilibrated in TBS buffer was added to the mixtures and incubated for 2 hr at 4°C with agitation. After that, resin was washed 4 times with 10 resin volumes of TBS buffer, and elution was performed by incubation for 30 min at room temperature with TBS buffer supplemented with either FLAG peptide (for FLAG-tagged proteins) or 3xFLAG peptide (for 3xFLAG-tagged proteins) at a final concentration of 0.15 µg/ml.

## Cryo-EM

3 µl of freshly purified RUVBL1-RUVBL2-DHX34 complex were applied to Quantifoil 300 mesh R1.2/1.3 grids after glow discharge, and the sample was flash frozen in liquid ethane using FEI Vitrobot MAG IV (Thermo Fisher Scientific). 3047 movies were collected in a Titan Krios at eBIC (Diamond Light Source, Oxford, UK) using a GATAN K2 Summit detector in counting mode, and a slit width of 20 eV on a GIF-Quantum energy filter. Cryo-EM images were collected as part of proposal EM20135 (Stop cancer - structural studies of macromolecular complexes involved in cancer by cryo-EM). Microscope calibrations and automatic data acquisition were performed with EPU software (Thermo Fisher Scientific) at a nominal magnification of ×47756, physical pixel size of 1.047 A°, a total dose of 48.1 $e^-/A°^2$, 6 $e^-/A°^2$/s, 40 fractions, and three images per hole. Autofocus was performed using an

objective defocus range between −1.5 and −3.0 µm. The oligomeric state of RUVBL1-RUVBL2 ATPase dead-mutants His-RUVBL1$^{E303Q}$-RUVBL2$^{E300Q}$ was analyzed by EM of negative stained samples. 3 µl microliters of freshly purified complexes (0.01 mg/ml) were deposited onto freshly glow-discharged carbon-coated 400 mesh copper electron microscopy (EM) grids (Electron Microscopy Sciences) and stained using 2% (w/v) uranyl acetate. Grids were visualized on a Tecnai 12 transmission electron microscope (Thermo Fisher Scientific Inc) with a lanthanum hexaboride cathode operated at 120 keV. Micrographs were recorded at x61320 nominal magnification and 2.5 A/pix on a 4k × 4 k TemCam-F416 CMOS camera (TVIPS).

## Image processing

MotionCor2 was used for local drift correction (5 × 5 patches) and dose-weighting of fraction stacks (*Zheng et al., 2017*). Parameters for the contrast transfer function (CTF) of drift-corrected images were determined with Gctf (*Zhang, 2016*). A subset of manually picked particles was used to generate reference-free 2D averages in RELION (*Zivanov et al., 2018*) that were further used for template-based automatic particle picking using Gautomatch (K. Zhang, Yale University). Initial data set containing 353 057 particles was 2D-classified in RELION, and best quality particles were selected (121449). Ab initio 3D model was generated from the selected particles using cryoSPARC (*Punjani et al., 2017*), and used as reference for 3D-classification in RELION-3 (*Zivanov et al., 2018*). After some 2D classification to select the most homogenous sub-class, image processing was divided in two branches (*Figure 2—figure supplement 1*). To determine the structure of the full complex, we first classified the images using a mask to remove those particles with density too small to account for DHX34, obtaining a data sub-set of 41237 particles. These were classified in 3D and refined in RELION to generate a cryo-EM map with an estimated average resolution of 5.0 Å using gold-standard methods and the Fourier Shell Correlation (FSC) cut-off of 0.143. Due to the large difference in resolution within the map, showing high resolution in the AAA-ring and lower resolution in DHX34, the final volume was sharpened using LocalDeblur, an automatic local resolution-based sharpening, as implemented in Scipion (*Ramírez-Aportela et al., 2020*).

In a parallel branch, we removed the influence of DHX34 during image processing by using a mask in subsequent rounds of 3D classification and refinement. Refinement of the RUVBL1-RUVBL2 was initiated from a preliminary refinement step (containing 101774 particles) and it was performed using a mask that excluded the density for DHX34 as well as the external OB-fold domains of RUVBL1-RUVBL2. An estimated average resolution of 4.2 Å was obtained for the RUVBL1-RUVBL2 using gold-standard FSC cut-off of 0.143. Local resolution distribution of the cryo-EM maps was estimated using ResMap (*Swint-Kruse and Brown, 2005*) as implemented in RELION-3.

To analyze the structure of DHX34 in the complex, the density was subtracted from the particles in the final map using a mask, as implemented in RELION-3 (*Zivanov et al., 2018*). Density subtracted particles were first subjected to a consensus refinement and further 3D classification. Significant heterogeneity was observed in 3D classified volumes, but these maps were not able to be refined due to the reduced number of particles. From the consensus refinement, a mask was applied to the volume for local refinement, obtaining a structure with an estimated average resolution of 10.4 Å using gold-standard FSC cut-off of 0.143.

Negative staining EM micrographs obtained for His-RUVBL1$^{E303Q}$-RUVBL2$^{E300Q}$ ATPase-dead mutants were used for extracting 55776 particles after CTF parameters estimation and correction, and 2D reference-free averages for each complex were obtained using *cis*TEM (*Grant et al., 2018*).

## Atomic model building

Atomic model building on the high-resolution cryo-EM maps of the RUVBL1-RUVBL2 ATPase ring was performed using the human RUVBL1-RUVBL2 crystal structure as starting model (PDB 2XSZ). After an initial rigid fitting in USCF Chimera, manual refinement for all the six chains was done using Coot (*Emsley et al., 2010*). The quality of the maps allowed the modeling of the majority of the side chains in the ATPase core of RUVBL1-RUVBL2 and the identification of the empty nucleotide pockets in all the three RUVBL2 present in the hexamer. Some regions of the DII domains were not as well defined in all the subunits, and we used the information of the three copies of each subunit in the ring to solve some of the ambiguities during modeling. A final step of automatic refinement was

done in phenix.real_space_refinement to improve the geometries of the model (*Afonine et al., 2018*).

## ATPase assays

ATP hydrolysis by His-RUVBL1-RUVBL2 was measured in a continuous spectrophotometric pyruvate kinase-lactate dehydrogenase-coupled assay, based on the regeneration of the hydrolyzed ATP coupled to oxidation of NADH (*Nørby, 1998*). NADH absorbance at 340 nm was measured using a Jasco V-550 UV/VIS Spectrophotometer with a Jasco EHC-477T Temperature Controller and monitored using the ND-1000 and Spectra Manager software in time course experiments, and its decrease was used to determine ATP hydrolysis rates. Assays were performed at 37°C in 100 μl reactions in buffer 50 mM Tris-HCl pH 7.4, 150 mM NaCl, 20 mM $MgCl_2$, containing 2 mM phosphoenolpyruvate (PEP), 0.5 mM NADH, 0.04 U/μl pyruvate kinase/0.05 U/μl lactic dehydrogenase (Sigma-Aldrich) and 5 mM ATP. The reaction components without the protein of study where incubated for 10 min until stabilization of the absorbance at 340 nm to allow the system to regenerate contaminant ADP. ATP hydrolysis reactions were started by addition of 3 μM of His-RUVBL1-RUVBL2 (concentration calculated considering monomers), either wild-type or ATPase-dead mutants, and in the absence or presence of 1 μM DHX34$^{D279A}$, and carried out for 20 min. We used the following ATPase-dead mutants: His-RUVBL1$^{E303Q}$-RUVBL2, His-RUVBL1-RUVBL2$^{E300Q}$ and His-RUVBL1$^{E303Q}$-RUVBL2$^{E300Q}$. A similar set of the experiments were performed with untagged RUVBL1-RUVBL2 complexes (wild-type and ATPase-dead mutants) after removal of the histidine tag present in RUVBL1. We also performed control experiments using wild-type DHX34 and the DHX34$^{D279A}$ mutant, and only buffer. Assays were performed at least by triplicate. ATP turnover (mol ATP/mol protein) indicated in $min^{-1}$ was calculated for a time interval during which the absorbance decrease was linear. Values in the graph are indicated as percentage of the rate of the wild-type protein.

## Thermal stability determination

Samples thermostability was measured by nano differential scanning fluorimetry using a Tycho NT.6 instrument (NanoTemper Technologies) that measures the intrinsic fluorescence from tryptophan and tyrosine residues detected at emission wavelengths of 350 and 330 nm, as a 30°C · min−one thermal ramp is applied from 35°C to 95°C. The observed changes in fluorescence signal allows to monitor the unfolding process of the protein. The temperature at which a transition occurs, the inflection temperature (*Ti*), was determined by detecting the maximum of the first derivative of the fluorescence ratios (F350/F330) after fitting experimental data with a polynomial function.

## Accession numbers

The cryo-EM maps of the RUVBL1-RUVBL2-DHX34 complex and the RUVBL1-RUVBL2 ring have been deposited in the EM database with accession codes EMD-11788 and EMD-11789 respectively. The structure of RUVBL1-RUVBL2 heterohexameric ring after binding of RNA helicase DHX34 has been deposited as PDB ID 7AHO.

## Acknowledgements

We acknowledge Diamond Light Source for access and support to the cryo-EM facilities at the UK's national Electron Bio-imaging Center (eBIC) under BAG proposal EM20135 (Stop cancer - structural studies of macromolecular complexes involved in cancer by cryo-EM), funded by the Wellcome Trust, MRC, and BBRSC. We acknowledge the use of the UCSF Chimera package from the Resource for Biocomputing, Visualization, and Informatics at the University of California, San Francisco. We thank Gianluca Degliesposti and Mark Skehel (MRC-LMB, UK) for preliminary analysis of the RUVBL1-RUVBL2-DHX34 complex using mass spectrometry. We thank Akio Yamashita and Shigeo Ohno from Yokohama City University (Japan) and Elena Conti from Max Planck Institute of Biochemistry (Germany, EU), for expression constructs and reagents. We also thank Dr Christos Savva from the Leicester Institute of Structural and Chemical Biology (UK), for initial data collection of the DHX34 complex. This work benefited from access to the Instruct Image Processing Center (I2PC, CNB-CSIC, Spain), an Instruct-ERIC centre, for image processing advice (PID: 10707).

## Additional information

### Funding

| Funder | Grant reference number | Author |
|---|---|---|
| Spanish Ministry of Science and Innovation | SAF2017-82632-P | Andrés López-Perrote<br>Carlos F Rodríguez<br>Marina Serna<br>Oscar Llorca |
| Autonomous Government of Madrid | Y2018/BIO4747 | Ana González-Corpas<br>Oscar Llorca |
| Autonomous Government of Madrid | P2018/NMT4443 | Ana González-Corpas<br>Oscar Llorca |
| MRC | Core funding | Javier F Caceres |
| Spanish Ministry of Science and Innovation | BES-2015-071348 | Carlos F Rodríguez |

The funders had no role in study design, data collection and interpretation, or the decision to submit the work for publication.

### Author contributions

Andres López-Perrote, Nele Hug, Ana González-Corpas, Carlos F Rodríguez, Marina Serna, Carmen García-Martín, Jasminka Boskovic, Rafael Fernandez-Leiro, Investigation; Javier F Caceres, Conceptualization, Writing - review and editing; Oscar Llorca, Conceptualization, Supervision, Funding acquisition, Writing - original draft, Project administration, Writing - review and editing

### Author ORCIDs

Andres López-Perrote (iD) https://orcid.org/0000-0001-7872-3442
Javier F Caceres (iD) http://orcid.org/0000-0001-8025-6169
Oscar Llorca (iD) https://orcid.org/0000-0001-5705-0699

### Decision letter and Author response

Decision letter https://doi.org/10.7554/eLife.63042.sa1
Author response https://doi.org/10.7554/eLife.63042.sa2

## Additional files

### Supplementary files

• Transparent reporting form

### Data availability

The cryo-EM maps of the RUVBL1-RUVBL2-DHX34 complex and the RUVBL1-RUVBL2 ring have been deposited in the EM database with accession codes EMD-11788 and EMD-11789 respectively. The structure of RUVBL1-RUVBL2 heterohexameric ring after binding of RNA helicase DHX34 has been deposited as PDB ID 7AHO.

The following datasets were generated:

| Author(s) | Year | Dataset title | Dataset URL | Database and Identifier |
|---|---|---|---|---|
| Lopez-Perrote A, Rodriguez CF, Llorca O | 2020 | RUVBL1-RUVBL2 heterohexameric ring after binding of RNA helicase DHX34 | https://www.rcsb.org/structure/7AHO | RCSB Protein Data Bank, 7AHO |
| López-Perrote A, Rodríguez CF, Llorca O | 2020 | Cryo-EM structure of the RUVBL1-RUVBL2-DHX34 complex | http://emsearch.rutgers.edu/atlas/11788_summary.html | EMDataBank, EMD-11788 |
| López-Perrote A, | 2020 | RUVBL1-RUVBL2 heterohexameric | http://emsearch.rutgers. | EMDataBank, EMD- |

| Rodríguez CF, Llorca O | ring after binding of RNA helicase DHX34 | edu/atlas/11789_summary.html | 11789 |
|---|---|---|---|

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
