## [Decision Letter]

**Acceptance summary:**

This study provides the first structural and biochemical evidence that RuvBL1/RuvBL2 directly interacts with the RNA helicase DHX34, suggesting a potential mechanism for the previously described activity of this AAA ATPase in the initiation of nonsense-mediated mRNA decay (NMD). The presented cryo-EM structure reveals how DHX34 binding to the RuvBL1/RuvBL2 heterohexamer induces a conformational change of RuvBL2's N-terminus and consequently modulates nucleotide binding and hydrolysis in every other subunit of the ATPase ring, potentially acting as a switch in orchestrating the assembly of NMD factors.

**Decision letter after peer review:**

[Editors’ note: the authors submitted for reconsideration following the decision after peer review. What follows is the decision letter after the first round of review.]

Thank you for submitting your work entitled "Regulation of RUVBL1-RUVBL2 AAA-ATPases by the nonsense-mediated mRNA decay factor DHX34, as evidenced by Cryo-EM" for consideration by *eLife*. Your article has been reviewed by three peer reviewers, including Andreas Martin as the Reviewing Editor and Reviewer #1, and the evaluation has been overseen by a Senior Editor.

As you will see in the individual reviews, there are a number of issues and weaknesses, both, for the structure determination and the biochemical characterization, and with several of them consistently picked up by multiple reviewers. Our decision has been reached after consultation between the reviewers, and based on these discussions and the individual reviews below, we regret to inform you that your work cannot be considered for publication in *eLife* at this point. However, reviewers agreed that the manuscript is potential interesting, and we invite you re-submit your manuscript after the critical corrections, analyses, and additional experiments have been completed.

Reviewer #1:

López-Perrote and colleagues present the structure of DHX34-bound RUVBL1-RUVBL2, which is implicated in nonsense-mediated mRNA decay (NMD). This structure demonstrates how DHX34 uses most of its domains to interact with the internal regions of the DII domains in both RUVBL1 and RUVBL2, and causes large conformational changes in the RUVBL1-RUVBL2 hexamer, in particular the ATPase domain of RUVBL2. DHX34 is identified as a potential regulator of RUVBL1-RUVBL2's ATPase activity, which represents an important step in determining the mechanisms underlying the initiation of NMD.

However, before this manuscript can be considered for publication in *eLife*, the authors should address several concerns, as outlined below.

Major Points:

1) The authors suggest that oligomerization of RUVBL1 and RUVBL2 hampers nucleotide exchange, yet this model seems not sufficiently supported by the data and the authors should adjust their discussion in this respect.

The ATP-binding pocket is located between neighboring protomers, with critical motifs contributed by both AAA domains, such that monomeric RUVBL1 and RUVBL2 are not capable of ATP hydrolysis. The comparison between hexamer and monomer is therefore unnecessary, and proposing that hexamerization lowers the ATP-hydrolysis rate does not make much sense. In fact, the subunit interface between RUVBL1 and RUVBL2 appears highly similar to that of other AAA+ motors, and it is in my opinion unlikely that the pocket itself "traps" the nucleotide and prevents exchange. At the various stages of the ATPase cycle (or positions in the hexameric ring), individual subunits of numerous other AAA+ hexamers show differential opening and closure of their nucleotide-binding sites. Depending on the averaging for the RUVBL1-RUVBL2 structure, subunits may appear uniformly closed, which, however, is not necessarily inconsistent with hydrolysis activity. What could contribute to the low rates, as suggested by the authors and previous studies, are the N-termini of RUVBL1/2 that seem to contact the nucleotide and in the case of RuvBL2 get released upon binding of DHX34. Given the minimal ATPase rate of 1 per min, RUVBL1/2 may indeed not work as a processive ATPase, but only as a switch that could get triggered by DHX34. In that respect this new structure is very interesting.

2) The overall analysis of nucleotide occupancies is problematic, considering that the resolution is not high enough to confidently assign nucleotides to each site.

The inherent averaging that occurs in single-particle cryo-EM image processing may very well explain the absence of detectable nucleotide in all RUVBL2 pockets of the DHX34-bound complex. As observed for many other cryo-EM structures of AAA+ motors, nucleotide pockets are highly dynamic and often less well resolved compared to the rest of the structure. Whether binding of DHX34 indeed induces a 3-fold symmetric state of the hexamer in which all 3 RUVBL2 sites are nucleotide free remains questionable. Building the class averages was likely determined by the asymmetric DHX34 density above the ring, and if the orientation of DHX34 is not well correlated with the nucleotide occupancy in a particular RuvBL2 subunit, averaging particles based on DHX34 could make it look like all 3 RuvBL2 sites are empty or lower in nucleotide density, whereas in reality it may be just one. If this were the case, the structure would resemble other AAA+ motors that show a small gap in the hexameric ring flanked by a nucleotide-free "seam" subunit, while all other subunits represent a continuum of nucleotide states.

To better evaluate the local resolution of the nucleotide pockets, it is necessary to have a zoomed-in view of these pockets on the ResMap of the 4.18Å RUVBL1-RUVBL2 hexameric ring (Figure 3—figure supplement 2D), highlighting the ADPs in RUVBL1. It is important to know the local resolution for these pockets, because based on the current evidence, the map alone may not provide enough detail to accurately assign nucleotides. One way to address this is by doing a more thorough analysis of the pockets themselves, including the overall size, shape, and location of key hydrolysis residues in comparison to known hydrolysis states of these pockets.

For example, in both Figure 4E and 4F it would be helpful to include labels of hydrolysis-relevant residues, like Arginine fingers and Walker A/B motifs, such that readers can easily orient themselves. In Figure 4E, the pocket looks just as "open" in the ATP-bound state (2XSZ) as in the DHX34-bound structure, and it is important to explain why this is considered a nucleotide-free rather than ADP- or ATP-bound state. Are there retracted residues that make this pocket incompetent for nucleotide binding? And could an ATP possibly fit into the RUVBL1 pocket? The current thresholding in Figure 4F does not exclude this possibility. It would also be appropriate to include the N-terminal histidines that directly interact with ADP in RUVBL1, as this may provide further evidence that it is indeed an ADP.

In general, it may be worth processing the data again and building class averages while masking out DHX34 to assess whether the hexamer indeed adopts a clear 3-fold symmetric state.

3) Regarding the very low ATPase rate of just 1 min-1, one potential issue may be that RUVBL1/2 was not purified in the presence of ATP and an ATP regeneration system. There are several examples of AAA+ motors that irreversibly lose robust activity when purified in the absence of ATP, and it may be worth testing whether RUVBL1/2 shows higher activities when purified in ATP.

The NADH consumption shown in Figure 5A is not linear, but increases over the 30 min measurements (30 – 60 min) for both, RUVBL1/2 and the DHX34-bound complex. What is the reason for that and what do the traces look like between the addition of RUVBL1/2 and the 30 min mark? The regeneration of ADP present in the RUVBL1/2 sample at the time of mixing should be completed within a couple of seconds, and temperature equilibration is expected to take only a couple of minutes. Non-linear absorbance changes over tens of minutes and a slow acceleration indicate that the system was not at steady state, which could also be consistent with the AAA+ motor being trapped in an inhibited state due to purification in the absence of ATP.

The authors discuss a model where ATP hydrolysis may regulate the interactions of RUVBL1/2 with other partners during NMD initiation, and the more stable binding of RUVBL1/2 to partners of the R2TP chaperone pathway in the absence of ATP hydrolysis is mentioned as an example. Similar effects have indeed been observed for various other AAA+ motors whose interactions are more dynamic during ATP hydrolysis. For RUVBL1/2, how does the very low ATPase activity of 1 min-1 compare to the off rate of its binding partners?

The authors propose that the ATPase inhibition of RUVBL1/2 by DHX34 may stabilize complexes. However, according to the presented model, DHX34 binding induces nucleotide release from every other site in the hexamer, which is expected to have distinct or even opposite effects compared to preventing hydrolysis and trapping hexamers in permanent ATP-bound states.

It is also suggested that DHX34 binding fully eliminates ATP hydrolysis (and even nucleotide interactions) in the RUVBL2 sites, while RUVBL1 "continues hydrolyzing at comparable levels to those measured in the absence of HBX34". This would mean that ATPase subunits in the hexamer are completely independent in their ATP hydrolysis, with no communication between neighbors. Although this is not ruled out, it has to my knowledge not been reliable described for other AAA+ hexamers, which usually show coordinated transitions and subunit communications that are mediated by arginine fingers and various other interactions within the topologically-closed rings. In fact, the ATPase rates for the single Walker-B mutants RUVBL1(E303Q)-RUVBL2 and RUVBL1-RUVBL2(E318Q) do not show 50% lower activity, but a reduction by 80 or 75% (Figure 5—figure supplement 1B) compared to wild type, suggesting that there is indeed communication between neighboring subunits.

The authors may consider further investigating this, for instance by characterizing hexamers with Walker-A or Walker-B mutations in RUVBL2, or an Arginine-finger mutation in RUVBL1 in the presence and absence of HDX34. If the authors' model is correct, the ATPase activity of these RUVBL1/2 variants should not respond to DHX34 binding and be similar to that of DHX34-bound wild-type RUVBL1/2.

Reviewer #2:

Lopez-Perrote et al. show that RUVBL1-RUVBL2 participates in the nonsense-mediated mRNA decay (NMD) pathway through direct interaction with the DHX34 RNA helicase. The authors present a cryo-EM structure of the complex, as well as pulldowns and functional assays that indicate DHX34 affects the conformation and activity of RUVBL1-RUVBL2.

Is there any indication of stoichiometry of DHX34 binding beside the triangular shape of the DHX34 density in the map (in Figure 3 and Figure 3—figure supplement 3)? The homology model fit into the map (in Figure 3—figure supplement 3) is unconvincing as there are clear helical densities in the map that appear not to fit any of the homology model helices. Overall, the homology model and experimental map do not appear to be in good agreement. Could more than one DHX34 be binding? The map and model in their current form do not seem sufficient to answer this question.

It seems surprising that the deletion of any of the domains of DHX34 (Figure 3—figure supplement 4C) results in no loss of binding to RUVBL1-RUVBL2. This observation is particularly surprising because it suggests that any domain can be deleted without affecting the folding or soluble expression of DHX34. It is not clear from this experiment that there is a definitive threshold for "loss of binding".

Further, the large variance in signal in the western blot appears to indicate that there could be a dependence on certain domains to bind (for instance RecA1), but the threshold for "no binding" is defined poorly. The authors should likely revise or modify the conclusion that this experiment supports the binding of all domains of DXH34 to RUVBL1-RUVBL2.

It is unclear if the 50% inhibition seen is due to incomplete binding of RUVBL1-RUVBL2 by DHX34 or if that 50% inhibition is an inherent property of the complex between the two.

Reviewer #3:

The work described in this manuscript is potentially of interest but is not ready for publication in its present form for a number of reasons. While it may be difficult (impossible?) for additional lab work to be conducted at present, this should not mean that incomplete studies are suitable for publication.

1) The experiments in Figure 1 Panel D are done by mixing components in solution and allowing them to come to some sort of equilibrium. This can lead to results that are not easy to interpret correctly in the absence of appropriate controls. For example, the amount of RvbL1/2 is not constant across the 2nd gel (compare lanes 10 and 11), which suggests a problem. If the whole RvbL/SMG1/RPAP3/PIH1D1 complex is unable to be bound to the resin, then an amount will remain in solution in samples with RPAP3/PIH1D1. This is not the same result as competing for sites on RvbLs, this is competition between Rvbs and RPAP3-PIH1D1 for a site or sites on SMG1-8-9. Alternatively, a complex between SMG complex or its components and RPAP3-PIH1D1 would not stick to the resin but might prevent binding of RvbLs to SMG1. Pull downs using the FLAG tag would provide a necessary control (i.e. repeating part (C) but in the presence of RPAP3-PIH1D1). However, it would also need to be shown that SMG8-9 does not interact with RPAP3-PIH1D1 as well.

Also, in the final lane (lane 14) all bands are more intense than even the 1:4 ratio lane (lane 13), which could be consistent with a portion of RPAP3/PIH1D1 remaining in solution (e.g. bound to SMG complex or a component of it) when the SMG complex is present. It also needs to be stated somewhere what the concentration on RPAP3-PIH1D1 is in lanes 2, 7, 9 and 14. From the gel band densities in the input it would appear to be 1:4.

Consequently, the data cannot distinguish between at least three different situations (a) competition between SMG1 and RPAP3-PIH1D1 for RvbL hexamer, (b) both binding simultaneously (as discussed by authors), or (c) binding of RPAP3-PIH1D1 to some component of SMG complex that then precludes binding of either to RvbL hexamer. These alternatives need to be distinguished for these data to be of any value.

In fact, since these data have no relevance to the rest of the paper they could be deleted. If they are included, then they need to be improved e.g. by cryoEM to show whether SMG1-Rvb complex is hexamer or dodecamer and/or where SMG1 is located. In their present form the data are not convincing without further validation and/or suitable controls.

2) For the cryoEM study, it is not clear to me why after the Rvb component was masked off so a 4.2Å structure could be obtained, this was not then used to subtract the Rvb density to allow a better local refinement of the DHX34 component? This could improve the DHX34 density dramatically. The observation that the Rvb hexamer density improves so much when the DHX34 component is removed, suggests that there is enough signal from that part to cause the misalignment of the RvbL hexamer so should be sufficient to allow refinement of that part alone, even if that requires several conformational classes to be defined.

3) For the ATPase inhibition experiments there are a number of issues.

First, why do the activity traces begin at 30mins rather than time zero? The rates should be shown from the start of ATP turnover, initiated by, for example, addition of ATP or magnesium after allowing an incubation period for components to form complexes if necessary.

Second, the rates are not linear but are curves. The whole point of the coupled assay is that the ATP is regenerated so remains at a constant level and therefore the rates should be linear unless other factors such as subunit association/disassociation are occurring that mean the system is not at equilibrium. Unless the rates are linear then they are meaningless because they are not steady state. Which part of these curves were measured to estimate the rates? The Materials and methods section suggests an amount after 30 mins was determined, presumably simply a difference over that time? Which time interval? Obviously, this is not accurate or appropriate for a rate that is curving. Interesting, in every assay shown, the curves are getting faster showing the rates are getting quicker as time progresses. This needs to be explained, particularly for DHX34.

Third, the experiments need to address whether it is the Vmax for the reaction that has altered or whether affinity for ATP is different. Furthermore, the structure raises the intriguing possibility that the rate may be halved because only half of the ATPase sites are now active i.e. those in the RvbL1 subunits. The authors have already created the tools to follow this up biochemically by making so-called Walker B mutants for each RvbL subunit. If it is indeed the RvbL2 subunits that are inactivated by the helicase binding, then binding should have no, or lesser, effect on the ATPase activity in the RvbL1/RvbL2EQ hexamer while the RvbL1EQ/RvbL2 hexamer should show a more dramatic effect than wildtype RvbL1/RvbL2 hexamers, or even complete inhibition of activity.

---

## [Author Response]

[Editors’ note: the authors resubmitted a revised version of the paper for consideration. What follows is the authors’ response to the first round of review.]

Reviewer #1:López-Perrote and colleagues present the structure of DHX34-bound RUVBL1-RUVBL2, which is implicated in nonsense-mediated mRNA decay (NMD). This structure demonstrates how DHX34 uses most of its domains to interact with the internal regions of the DII domains in both RUVBL1 and RUVBL2, and causes large conformational changes in the RUVBL1-RUVBL2 hexamer, in particular the ATPase domain of RUVBL2. DHX34 is identified as a potential regulator of RUVBL1-RUVBL2's ATPase activity, which represents an important step in determining the mechanisms underlying the initiation of NMD.However, before this manuscript can be considered for publication in eLife, the authors should address several concerns, as outlined below.Major Points:1) The authors suggest that oligomerization of RUVBL1 and RUVBL2 hampers nucleotide exchange, yet this model seems not sufficiently supported by the data and the authors should adjust their discussion in this respect.The ATP-binding pocket is located between neighboring protomers, with critical motifs contributed by both AAA domains, such that monomeric RUVBL1 and RUVBL2 are not capable of ATP hydrolysis. The comparison between hexamer and monomer is therefore unnecessary, and proposing that hexamerization lowers the ATP-hydrolysis rate does not make much sense. In fact, the subunit interface between RUVBL1 and RUVBL2 appears highly similar to that of other AAA+ motors, and it is in my opinion unlikely that the pocket itself "traps" the nucleotide and prevents exchange. At the various stages of the ATPase cycle (or positions in the hexameric ring), individual subunits of numerous other AAA+ hexamers show differential opening and closure of their nucleotide-binding sites. Depending on the averaging for the RUVBL1-RUVBL2 structure, subunits may appear uniformly closed, which, however, is not necessarily inconsistent with hydrolysis activity. What could contribute to the low rates, as suggested by the authors and previous studies, are the N-termini of RUVBL1/2 that seem to contact the nucleotide and in the case of RuvBL2 get released upon binding of DHX34.

The reviewer is right that our initial version of the manuscript did not clearly explain the consequences of RUVBL1 and RUVBL2 oligomerization. We would like to emphasize that our statement indicating that “*oligomerization of RUVBL1 and RUVBL2 hampers nucleotide exchange”* was not directly drawn from our data, but rather an elaboration based on previous results found in the literature

“*Crystal structure of the human AAA+ protein RuvBL1. Matias et al., 2006*”

Some relevant quotes from this paper stated

In an attempt to understand the weak ATPase activity of wild-type RuvBL1, we undertook a more detailed comparison between the nucleotide binding pockets of RuvBL1 and of other AAA^+^ proteins with known ATPase activity in vitro*; (…) The results of this comparison are listed in Table 3 and show that RuvBL1 has the lowest solvent-accessible area among all these molecules, indicating in this case a very tightly bound ADP unit. Therefore, it cannot easily exchange with ATP, and this may be the cause for the low* in vitro *ATPase activity of RuvBL1. In addition, the adenine ring of ADP is held in place by a large number of hydrogen bonds and hydrophobic contacts, and both phosphate groups also have a large number of hydrogen bonds.*Hexamer formation does not appear to influence ADP binding, since the capping of the nucleotide binding pocket by an adjacent monomer does not alter the solvent-accessible area calculations. However, it does obstruct a possible ADP exit channel and, thus, contributes to prevent the ADP/ATP Exchange.The three-dimensional structure of RuvBL1 reveals an ADP molecule tightly bound between DI and DIII and that access to the ATPase active site is additionally blocked by hexamerization, thereby making the exchange between ADP and ATP impossible. Additional cofactors are, therefore, likely to be needed to open the nucleotide pocket.

We agree with the reviewer that it is unlikely that the pocket itself could trap the nucleotide and it is probably that the N-termini of RUVBL1 and RUVBL2 regulates RUVBL1-RUVBL2 complexes.

Furthermore, our use of the terms “open” and “close” was not accurate, since we used it to refer to the absence of N-termini, where the access to the nucleotide binding pocket seems more exposed (“open”) compared to when the N-termini is shown in contact with the nucleotide (“closed”). We never intended to imply that the actual conformation of the nucleotide binding pockets was “open” or “closed”. We have now removed these terms and clarify this in the revised version.

Altogether, we have now extensively revised the appropriate Sections, mainly in the Introduction and in the Discussion.

Our revised conclusion is that DHX34 regulates hydrolysis by altering the N-termini of RUVBL2 that contact the nucleotide.

Given the minimal ATPase rate of 1 per min, RUVBL1/2 may indeed not work as a processive ATPase, but only as a switch that could get triggered by DHX34. In that respect this new structure is very interesting.

This is indeed interesting, and it is now mentioned in the Introduction and Discussion sections of the revised version. We have also commented on some recent manuscript that contains some new information about their rates of ATP hydrolysis (Nano et al., 2020).

2) The overall analysis of nucleotide occupancies is problematic, considering that the resolution is not high enough to confidently assign nucleotides to each site.The inherent averaging that occurs in single-particle cryo-EM image processing may very well explain the absence of detectable nucleotide in all RUVBL2 pockets of the DHX34-bound complex. As observed for many other cryo-EM structures of AAA+ motors, nucleotide pockets are highly dynamic and often less well resolved compared to the rest of the structure. Whether binding of DHX34 indeed induces a 3-fold symmetric state of the hexamer in which all 3 RUVBL2 sites are nucleotide free remains questionable. Building the class averages was likely determined by the asymmetric DHX34 density above the ring, and if the orientation of DHX34 is not well correlated with the nucleotide occupancy in a particular RuvBL2 subunit, averaging particles based on DHX34 could make it look like all 3 RuvBL2 sites are empty or lower in nucleotide density, whereas in reality it may be just one. If this were the case, the structure would resemble other AAA+ motors that show a small gap in the hexameric ring flanked by a nucleotide-free "seam" subunit, while all other subunits represent a continuum of nucleotide states.

We agree with this analysis made by the reviewer although we respectfully disagree we cannot confidentially assign nucleotides to each state. Some of the image processing suggested by the reviewer was actually done in the manuscript, which means that we did not explain well in the text and figures.

In the revised version we have clarified these key points but we have also performed additional experiments in accordance with the suggestions, to address these issues in full.

1) The reviewer mentions that the resolution is not sufficient to confidentially assign nucleotide.

We believe this not to be correct

a) Resolution at the nucleotide binding sites is around 4 Angstroms, which is sufficient to observe density for the nucleotides (see new Figure 3—figure supplement 1).

b) In this work, the structure of the RUVBL1-RUVBL2-DHX34 contains an internal control, since we can see nucleotides in RUVBL1 at the resolutions reached, and importantly the resolution is similar for both RUVBL1 and RUVBL2 subunits (see new Figure 3—figure supplement 1). Density for the nucleotide is clearly seen in RUVBL1 and clearly absent in RUVBL2, strongly supporting that there is nucleotide in all RUVBL1 subunits but not in RUVBL2 subunits.

Thus, in our opinion, the resolution of our map is sufficient to clearly detect nucleotides in RUVBL1 and we have a similar resolution in RUVBL2. (Issues about the influence of our image processing on the occupancy are mentioned below).

In the revised version we now show in detail the density in the cryo-EM map for the nucleotide-binding pockets of all 6 subunits of the complex showing clear density for the nucleotide in RUVBL1 subunits and clear absence of density in RUVBL2 subunits in new Figure 3—figure supplement 1.

2) Averaging may explain the absence of detectable nucleotide and class averages were determined by the asymmetric DHX34 density.

We agree with the reviewer that this can happen and this is why we used an image processing strategy designed to avoid it. We describe the structure of the full RUVBL1-RUVBL2-DHX34 complex in Figure 2, and then the structure of the RUVBL1-RUVBL2 ring at higher resolution in Figure 3. The structure of RUVBL1-RUVBL2 in Figure 3 was determined after masking out the density of DHX34 at very initial stages of refinement to remove any influence of DHX34 in the alignment and avoid the issues mentioned by the reviewer.

In summary, the reviewer is right, but we actually performed an image processing strategy to avoid the influence of DHX34 by masking out its density during refinement. Now we have clarified this in Results and Materials and methods section, but we have also modified the workflow of the image processing in the supplementary figures to clarify this (Figure 2—figure supplement 1).

3) New image processing and experiments to address this issue in full

We agree with the reviewer that it is critical to rule out that the lack of occupancy in all 3 RUVBL2 subunits is not an artifact of our image processing strategy.

Thus, in the revised version:

a) Additional image processing of the RUVBL1-RUVBL2-DHX34 complex:

After refinement of the RUVBL1-RUVBL2 ring masking out DHX34, we have now classified the particles in 3D to search for potential heterogeneity in nucleotide occupancy. As suggested above by the reviewer, we followed the state of each subunit by presence or absence of the histidine containing N-termini, a clear structural feature at this resolution level. This analysis, shown on Figure 3—figure supplement 3 did not detect any particles with different nucleotide occupancy.

b) Since the reviewer raised the issue that some of our conclusions could be influenced by the averaging of particles with a different level of nucleotide occupancy at the same location, we also analyzed the final RUVBL1-RUVBL2 structure by 3D classification of each of the 3 RUVBL1-RUVBL2 dimers in the structure independently. Each dimer in the final refinement step was classified without further alignment and masking out all other regions in the ring to search for heterogeneity. In all three dimers the vast majority of the particles displayed a clear conformation where the N-termini is present in RUVBL1 and absent in RUVBL2.

This analysis strongly suggested that for a great majority of particles, all RUVBL1-2 dimers have similar nucleotide occupancy. This analysis is now shown on Figure 3—figure supplement 4.

c) To further strengthen the results described in 3.2, we also analyzed our data using a symmetry expansion strategy as described before (Martino et al., 2018; Zhou et al., 2015). Briefly, because each RUVBL1-RUVBL2 complex has a roughly threefold symmetry we rotated each particle around the 3-fold symmetry axis three times to place all RUVBL1RUVBL2 dimers in the same position. This operation triplicated the data set and then we placed a mask around one of the dimers Particles were then subjected to a local classification strategy to look for heterogeneity in the nucleotide occupancy of each RUVBL1-RUVBL2 pair, regardless of each position in the ring. This is now shown in Figure 3—figure supplement 5, and confirms that the great majority of RUVBL1-RUVBL2 dimers have density for the N-terminus in RUVBL1 but not RUVBL2.

d) We have now performed ATPase experiments with several RUVBL1-RUVBL2 mutants that are compatible with an effect of DHX34 in all RUVBL2 subunits but not in RUVBL1 (see below and in response also to other reviewers)

Altogether, we believe that by clarifying the image processing performed by masking out DHX34 and all the new experiments described above, we have addressed the issue of the nucleotide occupancy in full.

To better evaluate the local resolution of the nucleotide pockets, it is necessary to have a zoomed-in view of these pockets on the ResMap of the 4.18Å RUVBL1-RUVBL2 hexameric ring (Figure 3—figure supplement 2D), highlighting the ADPs in RUVBL1. It is important to know the local resolution for these pockets, because based on the current evidence, the map alone may not provide enough detail to accurately assign nucleotides. One way to address this is by doing a more thorough analysis of the pockets themselves, including the overall size, shape, and location of key hydrolysis residues in comparison to known hydrolysis states of these pockets.

The reviewer is right. We have now modified Figure 3 to show details of the nucleotide binding pockets in RUVBL1 and RUVBL2 comparing our structure with previous crystal structures. As mentioned above, DHX34 affected mostly the N-termini of RUVBL2 but not significantly the nucleotide binding pockets themselves, as incorrectly implied in the previous version of the manuscript by using the terms “open” and “closed”. This has now been corrected.

Resolution at the nucleotide binding sites is around 4 Angstroms, which is sufficient to observe density for the nucleotides, and as suggested we now show the local resolution of the pockets themselves as well as detailed map in the new Figure 3—figure supplement 1.

For example, in both Figure 4E and 4F it would be helpful to include labels of hydrolysis-relevant residues, like Arginine fingers and Walker A/B motifs, such that readers can easily orient themselves. In Figure 4E, the pocket looks just as "open" in the ATP-bound state (2XSZ) as in the DHX34-bound structure, and it is important to explain why this is considered a nucleotide-free rather than ADP- or ATP-bound state. Are there retracted residues that make this pocket incompetent for nucleotide binding? And could an ATP possibly fit into the RUVBL1 pocket? The current thresholding in Figure 4F does not exclude this possibility. It would also be appropriate to include the N-terminal histidines that directly interact with ADP in RUVBL1, as this may provide further evidence that it is indeed an ADP.

The reviewer is right and all this information has now been included in the revised version, in the new Figure 3. We previously used the term “open” to refer to the disappearance of the N-termini of RUVBL2 that facilitates the accessibility to the nucleotide pockets. We now think that this term is not very adequate because it might be interpreted as implying that the nucleotide pocket itself was more open, which is not what happens. We have now modified the revised version and clarify our findings, and we also make a comparison of the nucleotide pockets of RUVBL1 and RUVBL2 before and after binding to DHX34.

In general, it may be worth processing the data again and building class averages while masking out DHX34 to assess whether the hexamer indeed adopts a clear 3-fold symmetric state.

As described above, we believe we have now addressed the issues regarding the occupancy if RUVBL1 and RUVBL2 subunits by nucleotide.

3) Regarding the very low ATPase rate of just 1 min-1, one potential issue may be that RUVBL1/2 was not purified in the presence of ATP and an ATP regeneration system. There are several examples of AAA+ motors that irreversibly lose robust activity when purified in the absence of ATP, and it may be worth testing whether RUVBL1/2 shows higher activities when purified in ATP.

The rates that we find for RUVBL1-RUVBL2 (4.9 mol ATP/mol protein·min) are in the same range that all previous works with these proteins, such as those found in Gorynia et al., 2011 (0.56 mol ATP/mol protein·min) and Lakomek et al., 2015 (3 mol ATP/mol protein·min). Interestingly, a recent work analyzed the ATPase activity of RUVBL1 and RUVBL2 separately, observing that the activity of RUVBL2 is 8-fold higher than RUVBL1 (Nano et al., 2020).

RUVBL1-RUVBL2 complexes co-purify with nucleotide regardless of these not been present during purification. This has been observed in all structures solved till now where nucleotide, mostly ADP, but sometimes also ATP, fill every subunit in the complex. We think, in agreement with all previous reports, that these ATPases have an intrinsic low ATPase rate.

The NADH consumption shown in Figure 5A is not linear, but increases over the 30 min measurements (30 – 60 min) for both, RUVBL1/2 and the DHX34-bound complex. What is the reason for that and what do the traces look like between the addition of RUVBL1/2 and the 30 min mark? The regeneration of ADP present in the RUVBL1/2 sample at the time of mixing should be completed within a couple of seconds, and temperature equilibration is expected to take only a couple of minutes. Non-linear absorbance changes over tens of minutes and a slow acceleration indicate that the system was not at steady state, which could also be consistent with the AAA+ motor being trapped in an inhibited state due to purification in the absence of ATP.

We apologize since we did not explain well the details of how we performed the ATPase experiments. The method we performed has been used by others to measure the activity of other ATPases and also of RUVBL1 and RUVBL2 (Nano et al., 2020). We have now clarified these issues.

We previously indicated that this experiment took place during 60 min in total, but it should be noted that the first 10 minutes were a pre-incubation time without the ATPase to allow the system to regenerate possible contaminant ADP present in the reagents, and not the reaction itself. We calculated rates of ATP consumption by averaging for a 30 min interval (from minute 30 to minute 60) during which the absorbance decrease was adjusted to a linear function.

In any case, we have decided to repeat and improve these experiments, at the same time that some mutants were also analyzed.

For this revised version:

– In our curves, linearity was lost only at the end experiment at 37ºC. We suspect that this could be due to the instability of DHX34. In our hands, DHX34 activity is affected over time upon storage and it might be possible that it is also affected after a long period at 37º. To investigate this, we have now analyzed the stability of DHX34 using nano differential scanning fluorimetry, a technique that measures the intrinsic fluorescence of the protein during a thermal ramp denaturation experiment. We found that DHX34 is stable during 20 min incubation at 37ºC but not at longer times (Figure 5—figure supplement 1), so accordingly experiments have been run for only 20 min.

– We have repeated the experiments for 20 min and NADH consumptions are now linear (Figure 5—figure supplement 2).

– We have clarified our methodology in Materials and methods.

The authors discuss a model where ATP hydrolysis may regulate the interactions of RUVBL1/2 with other partners during NMD initiation, and the more stable binding of RUVBL1/2 to partners of the R2TP chaperone pathway in the absence of ATP hydrolysis is mentioned as an example. Similar effects have indeed been observed for various other AAA+ motors whose interactions are more dynamic during ATP hydrolysis. For RUVBL1/2, how does the very low ATPase activity of 1 min-1 compare to the off rate of its binding partners?

As mentioned, current evidences suggest that the low ATPase activity of RUVBL1-RUVBL2 could be related to its function as a switch in different macromolecular complexes, but not as a processive motor ATPase. In several pathways, such as NMD, protein-protein interactions are transient due the dynamics of the process, and signals from others partners are needed for rearrangement steps to allow the remodeling of the complexes. RUVBL1-RUVBL2 ATPase activity could have an impact on such signals, allowing assembly and/or disassembly of intermediate complexes.

The authors propose that the ATPase inhibition of RUVBL1/2 by DHX34 may stabilize complexes. However, according to the presented model, DHX34 binding induces nucleotide release from every other site in the hexamer, which is expected to have distinct or even opposite effects compared to preventing hydrolysis and trapping hexamers in permanent ATP-bound states.

At this stage we do not know what the function of the regulation of RUVBL1-RUVBL2 activity by DHX34 in the context of NMD is. Our speculation is based on recent findings showing that an allosteric inhibitor of RUVBL1-RUVBL2 ATPase activity stabilizes their interaction with clients of the PIKK assembly pathway in cells. Since DHX34 reduces hydrolysis, this could have a similar effect as the inhibitor described. Having said that, this is very speculative and we modified the Discussion section to give less strength to this educated guess.

It is also suggested that DHX34 binding fully eliminates ATP hydrolysis (and even nucleotide interactions) in the RUVBL2 sites, while RUVBL1 "continues hydrolyzing at comparable levels to those measured in the absence of HBX34". This would mean that ATPase subunits in the hexamer are completely independent in their ATP hydrolysis, with no communication between neighbors. Although this is not ruled out, it has to my knowledge not been reliable described for other AAA+ hexamers, which usually show coordinated transitions and subunit communications that are mediated by arginine fingers and various other interactions within the topologically-closed rings. In fact, the ATPase rates for the single Walker-B mutants RUVBL1(E303Q)-RUVBL2 and RUVBL1-RUVBL2(E318Q) do not show 50% lower activity, but a reduction by 80 or 75% (Figure 5—figure supplement 1B) compared to wild type, suggesting that there is indeed communication between neighboring subunits.The authors may consider further investigating this, for instance by characterizing hexamers with Walker-A or Walker-B mutations in RUVBL2, or an Arginine-finger mutation in RUVBL1 in the presence and absence of HDX34. If the authors' model is correct, the ATPase activity of these RUVBL1/2 variants should not respond to DHX34 binding and be similar to that of DHX34-bound wild-type RUVBL1/2.

We agree with the reviewer that we cannot conclude that DHX34 affects only RUVBL2 sites with the data available in our first version of the manuscript. To make things more complex, recent evidence shows that RUVBL2 subunits might be more active than RUVBL1 subunits (Nano et al., 2020). Therefore, calculations on the ratio of reduction of activity after adding DHX34 are not sufficient to relate the observed phenomena to either RUVBL1 or RUVBL2 subunits.

To address this issue, we have performed the experiments suggested by the reviewer which are part of the new Figure 5 and its supplemental figures:

– We generated RUVBL1 and RUVBL2 double mutants unable to bind nucleotide, and we used them to purify RUVBL1-RUVBL2 complexes with one or both subunits mutated. As a control, we analyzed the complex by electron microscopy before performing experiments (Figure 5—figure supplement 1A, B). To our surprise, we were unable to find the well-characterized hexameric top views, and complexes appeared as heterogeneous in size and shape. Other groups have found before that ADP or ATP could help the assembly of recombinant RUVBL2, and we interpreted that assembly of recombinant hexameric complexes is affected when nucleotide cannot bind to both subunits.

– We then focused on RUVBL1 and RUVBL2 mutations that are affected in ATP hydrolysis but not ATP binding. Complexes containing one or both subunits mutated were purified and their correct assembly verified using electron microscopy.

– We also tested the ATPase activity of the different mutants in the absence and presence of DHX34. This information is now found in Figure 5 and fully confirms that DHX34 affects only RUVBL2, since DHX34 effects are only detected in complexes containing wildtype RUVBL2 but not ATPase-dead RUVBL2 mutants.

A key point to our work was that the effects of DHX34 on ATP hydrolysis correlate well with the conformational changes observed in RUVBL2 but not RUVBL1 subunits. The new experiments added a strong support this conclusion.

Reviewer #2:Lopez-Perrote et al. show that RUVBL1-RUVBL2 participates in the nonsense-mediated mRNA decay (NMD) pathway through direct interaction with the DHX34 RNA helicase. The authors present a cryo-EM structure of the complex, as well as pulldowns and functional assays that indicate DHX34 affects the conformation and activity of RUVBL1-RUVBL2.Is there any indication of stoichiometry of DHX34 binding beside the triangular shape of the DHX34 density in the map (in Figure 3 and Figure 3—figure supplement 3)? The homology model fit into the map (in Figure 3—figure supplement 3) is unconvincing as there are clear helical densities in the map that appear not to fit any of the homology model helices. Overall, the homology model and experimental map do not appear to be in good agreement. Could more than one DHX34 be binding? The map and model in their current form do not seem sufficient to answer this question.

The reviewer is right that the homology model does not fit the structure of the complex, and we indicated this in the text. We have removed this fitting experiment in the manuscript since, we agree with the reviewer, it does not provide a sufficiently useful information.

To determine the stoichiometry of the RUVBL1-RUVBL2 complex:

1) We have extracted the densities for DHX34 from the map of the complex. These images were processed to obtain a cryo-EM map of DHX34 processed independently of RUVBL1-RUVBL2. Although resolution was poor given the limited number of images, insufficient for such a small and flexible protein, this map was useful to compare with the structure of isolated DHX34 from negative stain microscopy. This comparison suggested that only one molecule of DHX34 was bound to RUVBL1-RUVBL2.

2) In addition, we estimated the mass of this volume as 120 kDa using the “volume” programme from EMAN (see text for details).

Together these results indicate that only one DHX34 binds to each RUVBL1-RUVBL2 and that we only probably visualize the core of the protein and not the flexible C-terminal domain. Details can be found in the final section of Results and in the new Figure 4—figure supplement 1

It seems surprising that the deletion of any of the domains of DHX34 (Figure 3—figure supplement 4C) results in no loss of binding to RUVBL1-RUVBL2. This observation is particularly surprising because it suggests that any domain can be deleted without affecting the folding or soluble expression of DHX34. It is not clear from this experiment that there is a definitive threshold for "loss of binding".Further, the large variance in signal in the western blot appears to indicate that there could be a dependence on certain domains to bind (for instance RecA1), but the threshold for "no binding" is defined poorly. The authors should likely revise or modify the conclusion that this experiment supports the binding of all domains of DXH34 to RUVBL1-RUVBL2.

The DHX34 mutants we used were designed and selected by our collaborators in Edinburgh (Nele Hug and Javier F Caceres). In our previous collaborative study, we showed that they soluble and maintain certain functional properties (Melero et al., 2016). Here, we made use of these already tested mutants to analyze their effect on DHX34 binding.

We agree with the reviewer that it is surprising that we do not have larger effects in some mutants. This could be partially due to the difficulty to confidentially detect small differences but maybe also to the possible influence of additional factors for the RUVBL1-RUVBL2-DHX34 interaction in cells, such as the contribution of additional NMD proteins.

We have now repeated these experiments and quantified the effect of individual domains of DHX34 and larger deletions covering several domains in the binding to RUVBL1-RUVBL2 (new Figure 4).

It is unclear if the 50% inhibition seen is due to incomplete binding of RUVBL1-RUVBL2 by DHX34 or if that 50% inhibition is an inherent property of the complex between the two.

The reviewer is right. In addition, recent evidence shows that RUVBL2 subunits might be more active than RUVBL1 subunits (Nano et al., 2020). Therefore, calculations on the ratio of reduction of activity after adding DHX34 are not sufficient to relate the observed phenomena to either RUVBL1 or RUVBL2 subunits.

To address this issue, we have made used of RUVBL1 and RUVBL2 mutant unable to hydrolysis ATP, and measure the influence of adding DHX34. Complexes containing one or both subunits mutated were purified and their correct assembly verified using electron microscopy. We then tested the ATPase activity of the different mutants in the absence and presence of DHX34.This information is now found in Figure 5 and its supplemented figures and fully confirms that DHX34 affects only RUVBL2, since DHX34 effects are only detected in complexes containing wildtype RUVBL2 but not ATPase-dead RUVBL2 mutants.

A key point to our work was that the effects of DHX34 on ATP hydrolysis correlate with the conformational changes observed in RUVBL2 but not RUVBL1 subunits. The new experiments added in Figure 5 strongly support this conclusion.

Reviewer #3:The work described in this manuscript is potentially of interest but is not ready for publication in its present form for a number of reasons. While it may be difficult (impossible?) for additional lab work to be conducted at present, this should not mean that incomplete studies are suitable for publication.

We agree with this reviewer and we have followed the criticisms and concerns raised by all three reviewers to be able to present a more complete study. Despite the very challenging situation due to the current pandemic, we believe that we have managed to do this, and we have revised our manuscript extensively, incorporated new data, and also revised the text. We believe that the revised version of the manuscript has addressed the issues raised by the reviewers.

1) The experiments in Figure 1 Panel D are done by mixing components in solution and allowing them to come to some sort of equilibrium. This can lead to results that are not easy to interpret correctly in the absence of appropriate controls. For example, the amount of RvbL1/2 is not constant across the 2nd gel (compare lanes 10 and 11), which suggests a problem. If the whole RvbL/SMG1/RPAP3/PIH1D1 complex is unable to be bound to the resin, then an amount will remain in solution in samples with RPAP3/PIH1D1. This is not the same result as competing for sites on RvbLs, this is competition between Rvbs and RPAP3-PIH1D1 for a site or sites on SMG1-8-9. Alternatively, a complex between SMG complex or its components and RPAP3-PIH1D1 would not stick to the resin but might prevent binding of RvbLs to SMG1. Pull downs using the FLAG tag would provide a necessary control (i.e. repeating part (C) but in the presence of RPAP3-PIH1D1). However, it would also need to be shown that SMG8-9 does not interact with RPAP3-PIH1D1 as well.Also, in the final lane (lane 14) all bands are more intense than even the 1:4 ratio lane (lane 13), which could be consistent with a portion of RPAP3/PIH1D1 remaining in solution (e.g. bound to SMG complex or a component of it) when the SMG complex is present. It also needs to be stated somewhere what the concentration on RPAP3-PIH1D1 is in lanes 2, 7, 9 and 14. From the gel band densities in the input it would appear to be 1:4.Consequently, the data cannot distinguish between at least three different situations (a) competition between SMG1 and RPAP3-PIH1D1 for RvbL hexamer, (b) both binding simultaneously (as discussed by authors), or (c) binding of RPAP3-PIH1D1 to some component of SMG complex that then precludes binding of either to RvbL hexamer. These alternatives need to be distinguished for these data to be of any value.In fact, since these data have no relevance to the rest of the paper they could be deleted. If they are included, then they need to be improved e.g. by cryoEM to show whether SMG1-Rvb complex is hexamer or dodecamer and/or where SMG1 is located. In their present form the data are not convincing without further validation and/or suitable controls.

We agree with the reviewer that the SMG1 interaction data is disconnected from the rest of the manuscript, which is focused on the interaction between RUVBL1-RUVBL2 and DHX34. We have now removed this from the revised version, including the competition experiments using RPAP3PIH1D1, which were a concern for this reviewer.

Nonetheless, we have left the main RUVBL1-RUVBL2-SMG1 interaction experiment as part of supplemental information in the context of the interaction experiments of RUVBL1-RUVBL2 with most factors involved in NMD initiation.

2) For the cryoEM study, it is not clear to me why after the Rvb component was masked off so a 4.2Å structure could be obtained, this was not then used to subtract the Rvb density to allow a better local refinement of the DHX34 component? This could improve the DHX34 density dramatically. The observation that the Rvb hexamer density improves so much when the DHX34 component is removed, suggests that there is enough signal from that part to cause the misalignment of the RvbL hexamer so should be sufficient to allow refinement of that part alone, even if that requires several conformational classes to be defined.

We have tried again to improve the structure of DHX34 as suggested (Figure 4—figure supplement 1), but without success, possibly because the number of particles is not sufficiently large for a small protein. All these new analyses are part of Figure 4—figure supplement 1 and one section of results. In addition, we have compared our low-resolution map of DHX34 when bound to RUVBL1-RUVBL2 with the low-resolution map of DHX34 obtained in isolation. This, and other analysis, suggests that the stoichiometry of the RUVBL1-RUVBL2-DHX34 complex is 3:3:1.

3) For the ATPase inhibition experiments there are a number of issues.

Some of the issues about the ATPase experiments mentioned by the reviewer have their origin in our lack of adequate explanations on how the experiments were conducted, and how our results were represented. We have now addressed this in full by providing significant more details in the Materials and methods section and by relating our experiments to a recent work that measures the ATPase activity of RUVBL1 and RUVBL2 using the same protocols we use.

First, why do the activity traces begin at 30mins rather than time zero? The rates should be shown from the start of ATP turnover, initiated by, for example, addition of ATP or magnesium after allowing an incubation period for components to form complexes if necessary.Second, the rates are not linear but are curves. The whole point of the coupled assay is that the ATP is regenerated so remains at a constant level and therefore the rates should be linear unless other factors such as subunit association/disassociation are occurring that mean the system is not at equilibrium. Unless the rates are linear then they are meaningless because they are not steady state. Which part of these curves were measured to estimate the rates? The Materials and methods section suggests an amount after 30 mins was determined, presumably simply a difference over that time? Which time interval? Obviously, this is not accurate or appropriate for a rate that is curving. Interesting, in every assay shown, the curves are getting faster showing the rates are getting quicker as time progresses. This needs to be explained, particularly for DHX34.

In our experiments, reaction components of the pyruvate kinase-lactate dehydrogenase-coupled assay and the RUVBL1-RUVBL2 and DHX34 protein complexes were incubated separately for 10 min at 37 ºC for stabilization. This step was done to allow the system to regenerate possible contaminant ADP from the reagents, so the ATPase reaction starts from a stable baseline. Then, the RUVBL1-RUVBL2 and DHX34 protein are added to the reaction to initiate hydrolysis, and the measurements were taken for additionally 50 min. The experiment took place during 60 min in total, but the first 10 minutes were the pre-incubation time, not the ATP hydrolysis reaction per se. This was not properly explained in the previous version of the manuscript. We calculated rates of ATP consumption by averaging for a 30 min interval during which the absorbance decrease was adjusted to a linear function (from minute 30 to minute 60). Similar approach has been used before (Nano et al., 2020). But the selection of the interval was arbitrary as long as rates were linear and we decided then to represent in the graphs maintaining the original numbering of minutes in our experiment. We have now repeated all the ATPase experiments in the manuscript, and in addition to new experiments using several RUVBL1 and RUVBL2 mutants. We have now clarified how the experiment was performed, and we now consider as time zero when the mix was performed but removing the first minute where the measurement is distorted by the opening and closing of the Jasco V-550 UV/VIS Spectrophotometer.

For this revised version:

– In our curves, linearity was lost only at the end experiment at 37ºC. We suspect that this could be due to the instability of DHX34. In our hands, DHX34 activity is affected over time upon storage and it might be possible that it is also affected after a long period at 37º. To investigate this, we have now analyzed the stability of DHX34 using nano differential scanning fluorimetry, a technique that measures the intrinsic fluorescence of the protein during a thermal ramp denaturation experiment, and we found that DHX34 is stable during 20 min incubation at 37ºC but not at longer times, so experiments have been run for only 20 min (Figure 5—figure supplement 1).

– We have repeated the experiments for 20 min after the mixing and NADH consumptions are now linear (Figure 5—figure supplement 2).

– We have clarified our methodology in the Materials and methods section.

All these changes are now part of Figure 5 and its supplemental figures and a more detailed version of Materials and methods.

Third, the experiments need to address whether it is the Vmax for the reaction that has altered or whether affinity for ATP is different. Furthermore, the structure raises the intriguing possibility that the rate may be halved because only half of the ATPase sites are now active i.e. those in the RvbL1 subunits. The authors have already created the tools to follow this up biochemically by making so-called Walker B mutants for each RvbL subunit. If it is indeed the RvbL2 subunits that are inactivated by the helicase binding, then binding should have no, or lesser, effect on the ATPase activity in the RvbL1/RvbL2EQ hexamer while the RvbL1EQ/RvbL2 hexamer should show a more dramatic effect than wildtype RvbL1/RvbL2 hexamers, or even complete inhibition of activity.

Recent evidence shows that RUVBL2 subunits might be more active than RUVBL1 subunits (Nano et al., 2020). Therefore, calculations on the ratio of reduction of activity after adding DHX34 are not sufficient to relate the observed phenomena to either RUVBL1 or RUVBL2 subunits. To address this issue, we have made use of RUVBL1 and RUVBL2 mutant unable to hydrolysis ATP, and have measured the influence of adding DHX34. Complexes containing one or both subunits mutated were purified and their correct assembly verified using electron microscopy. We then tested the ATPase activity of the different mutants in the absence and presence of DHX34. This information is now found in Figure 5 and full confirms that DHX34 affects only RUVBL2, since DHX34 effects are only detected in complexes containing wildtype RUVBL2 but not ATPase-dead RUVBL2 mutants.

A key point to our work was that the effects of DHX34 on ATP hydrolysis correlate with the conformational changes observed in RUVBL2 but not RUVBL1 subunits. The new experiments added strongly support this conclusion.